# Bone marrow Adipoq-lineage progenitors are a major cellular source of M-CSF that dominates bone marrow macrophage development, osteoclastogenesis, and bone mass

Kazuki Inoue[1,2†], Yongli Qin[1,2†], Yuhan Xia[1,2†], Jie Han[3], Ruoxi Yuan[1,2], Jun Sun[4], Ren Xu[3], Jean X Jiang[5], Matthew B Greenblatt[4,6], Baohong Zhao[1,2,7]*

[1]Arthritis and Tissue Degeneration Program and David Z. Rosensweig Genomics Research Center, Hospital for Special Surgery, New York, United States; [2]Department of Medicine, Weill Cornell Medical College, New York, United States; [3]The first Affiliated Hospital of Xiamen University-ICMRS Collaborating Center for Skeletal Stem Cells, State Key Laboratory of Cellular Stress Biology, Faculty of Medicine and Life Sciences, Fujian Provincial Key Laboratory of Organ and Tissue Regeneration, School of Medicine, Xiamen University, Xiamen, China; [4]Pathology and Laboratory Medicine, Weill Cornell Medical College, New York, United States; [5]Department of Biochemistry & Structural Biology, University of Texas Health Science Center at San Antonio, San Antonio, United States; [6]Research Institute, Hospital for Special Surgery, New York, United States; [7]Graduate Program in Cell and Development Biology, Weill Cornell Graduate School of Medical Sciences, New York, United States

*For correspondence:
zhaob@hss.edu

†These authors contributed equally to this work

**Abstract** M-CSF is a critical growth factor for myeloid lineage cells, including monocytes, macrophages, and osteoclasts. Tissue-resident macrophages in most organs rely on local M-CSF. However, it is unclear what specific cells in the bone marrow produce M-CSF to maintain myeloid homeostasis. Here, we found that Adipoq-lineage progenitors but not mature adipocytes in bone marrow or in peripheral adipose tissue, are a major cellular source of M-CSF, with these Adipoq-lineage progenitors producing M-CSF at levels much higher than those produced by osteoblast lineage cells. The Adipoq-lineage progenitors with high CSF1 expression also exist in human bone marrow. Deficiency of M-CSF in bone marrow Adipoq-lineage progenitors drastically reduces the generation of bone marrow macrophages and osteoclasts, leading to severe osteopetrosis in mice. Furthermore, the osteoporosis in ovariectomized mice can be significantly alleviated by the absence of M-CSF in bone marrow Adipoq-lineage progenitors. Our findings identify bone marrow Adipoq-lineage progenitors as a major cellular source of M-CSF in bone marrow and reveal their crucial contribution to bone marrow macrophage development, osteoclastogenesis, bone homeostasis, and pathological bone loss.

## Editor's evaluation

This fundamental work advances our understanding of the function of a subpopulation of bone marrow cells as an important source of M-CSF to regulate bone remodeling. The evidence supporting the conclusion is compelling, using Adipoq-Cre-driven conditional deletion of Csf1 and the analysis of the publicly available scRNAseq data. This paper is of interest to skeletal biologists studying bone marrow stem/progenitor cells and bone remodeling.

## Introduction

Macrophage colony-stimulating factor (M-CSF), encoded by the *Csf1* gene, is a growth factor that plays a crucial role in the proliferation, differentiation, survival and function of myeloid lineage cells, including monocytes, macrophages, and osteoclasts (*Stanley et al., 1997*; *Ross, 2006*). The global absence of *Csf1* in the *Csf1^op^/ Csf1^op^* mice, which carry an inactivating mutation in the coding region of *Csf1*, leads to tissue macrophage deficiency, growth retardation and skeletal abnormality. The severe osteopetrosis, shortened long bones and failure of dental eruption in *Csf1^op^/ Csf1^op^* mice attest to the essential role of M-CSF for osteoclast generation (*Cecchini et al., 1994*; *Dai et al., 2002*; *Felix et al., 1990*; *Wiktor-Jedrzejczak et al., 1982*; *Wiktor-Jedrzejczak et al., 1990*; *Yoshida et al., 1990*).

Osteoclasts are giant multinucleated cells responsible for bone resorption. They are derived from the monocyte/macrophage lineage and are a specialized terminally differentiated macrophage. Osteoclasts play an important role not only in physiological bone development and remodeling, but also actively contribute to musculoskeletal tissue damage and accelerating the progression of post-menopausal osteoporosis and inflammatory arthritis. Osteoclasts and their progenitors express the M-CSF receptor c-Fms. M-CSF is essential for the entire process of osteoclast differentiation, from the generation of osteoclast precursors to the formation and survival of mature osteoclasts (*Ross, 2006*; *Tanaka et al., 1993*; *Feng and Teitelbaum, 2013*). M-CSF also acts together with β3 integrin to regulate actin remodeling in osteoclasts to enable osteoclasts to spread, migrate, fuse, and form actin rings to facilitate bone resorption (*Ross, 2006*; *Ross and Teitelbaum, 2005*; *Insogna et al., 1997*). These findings underscore the indispensable role of *Csf1* in not only monocyte to macrophage development, but also osteoclast differentiation, function and survival, which directly influences bone mass.

M-CSF is expressed by a variety of cells, such as endothelial cells, myoblasts, epithelial cells, and fibroblasts (*Cecchini et al., 1994*; *Schrader et al., 1991*; *Clinton et al., 1992*; *Fibbe et al., 1988*). This diversity of cellular sources of M-CSF is likely due to the need to support a widely distributed network of tissue-resident macrophages, which require M-CSF for their development, function, and homeostasis. Under physiologic conditions, circulating M-CSF is mainly produced by vascular endothelial cells (*Clinton et al., 1992*; *Roth and Stanley, 1992*; *Ryan et al., 2001*). The role of circulating M-CSF in tissue macrophage support appears to be highly context dependent. For example, macrophages in kidney and liver are highly dependent on circulating M-CSF. Bone marrow resident macrophages, however, appear to not require circulating M-CSF, indicating the importance of local M-CSF produced in bone marrow (*Cecchini et al., 1994*). Osteocytes embedded in bone express M-CSF that contributes to bone mass maintenance (*Werner et al., 2020*). Although in vitro studies show that M-CSF is expressed by an osteoblast cell line, calvarial osteoblastic cells and a cloned bone marrow stromal cell line (*Tanaka et al., 1993*; *Elford et al., 1987*; *Zhao et al., 2002*; *Naparstek et al., 1986*), there is no clear evidence supporting whether osteoblasts express M-CSF in vivo and the contributions of osteoblast-derived M-CSF to macrophage and bone homeostasis in vivo is unclear. Therefore, it is largely unknown what specific bone marrow cellular population mainly produces this critical cytokine.

The rapid evolution of scRNAseq technology provides an opportunity to investigate transcriptomics at the individual cell level. In this study, we utilized bone marrow scRNAseq datasets (*Zhong et al., 2020*; *Wolock et al., 2019*; *Tikhonova et al., 2020*; *Baryawno et al., 2019*; *Baccin et al., 2020*; *Dolgalev and Tikhonova, 2021*), and identified a group of unique bone marrow cells featuring Adipoq expression that highly express *Csf1*. The level of *Csf1* expressed by these bone marrow Adipoq-lineage progenitor cells is much higher than that produced by osteoblast lineage cells. We further demonstrated the functional importance of the *Csf1* expressed by bone marrow Adipoq-lineage progenitors in macrophage development, osteoclastogenesis and bone mass maintenance.

## Results

### scRNAseq reveals Adipoq-lineage progenitors as a main cellular source expressing *Csf1* in bone marrow

Along with the rapid progress of single-cell RNAseq technology, single-cell transcriptomics provides an unprecedented assessment of tissue cellular composition and gene expression profile at individual cell resolution. We took advantage of a recently published dataset (*Dolgalev and Tikhonova, 2021*), which integrated three bone marrow scRNAseq datasets (*Tikhonova et al., 2020*; *Baryawno et al., 2019*; *Baccin et al., 2020*), and analyzed the expression profiles of non-hematopoietic bone marrow

cells. *Adipoq* is found to be most highly expressed in the cluster MSPC-adipo (mesenchymal progenitor cells-adipo lineage)/Adipoq-lineage progenitors (*Figure 1A, B and D*) as a marker of this cell population among the clusters. This MSPC-Adipo cluster (*Dolgalev and Tikhonova, 2021*) was described as the adipo-primed mesenchymal progenitors (*Tikhonova et al., 2020*), adipocyte progenitors (*Wolock et al., 2019*), Adipo-CAR (Cxcl12-Abundant Reticular) cells (*Baccin et al., 2020*), Lepr-MSC (*Baryawno et al., 2019*), or marrow adipogenic lineage precursors (MALPs) (*Zhong et al., 2020*) identified in bone marrow. Since we utilized Adipoq Cre mice to investigate the function of this progenitor population, we used the nomenclature bone marrow Adipoq-lineage progenitors to designate these cells throughout this study. We found the Adipoq-lineage progenitors to be highly enriched for bone marrow stromal cell markers important for the hematopoietic niche, such as *Lepr*, *Cxcl12* and *Kitl*, but also express unique genes, such as *Lpl* (*Figure 1B*, Figure 3A). These cells labeled by Adipoq Cre were relatively quiescent, with about 4% cells incorporating BrdU under basal conditions (*Figure 1C*). The Adipoq-lineage progenitors are not mature adipocytes, but express some common adipocyte lineage markers, such as *Cebpa* and *Adipoq* (*Figure 1B*). On the other hand, osteoblast lineage marker genes, such as *Sp7*, *Alpl*, *Dmp1,* and *Bglap*, are nearly undetectable in these cells (*Figure 1B*).

When screening expression of known genes regulating skeletal homeostasis in the bone marrow, we found that these bone marrow Adipoq-lineage progenitors (MSPC-adipo cluster) express the highest level of *Csf1* (*Figure 1D, E and F*), which was unexpected because the MSPC-osteo (mesenchymal progenitor cells-osteo lineage) cluster and osteoblasts were thought to be the cellular source expressing *Csf1* in bone marrow based on previous studies of in vitro cultured osteoblast cell lines or calvarial osteoblastic cells (*Tanaka et al., 1993*; *Elford et al., 1987*). Further quantitative analysis showed that bone marrow Adipoq-lineage progenitors (70% of 6441 cells expressed with scaled average expression level at 2.6) express a markedly higher level of *Csf1* than MSPC-osteo cluster cells (61% of 2247 cells expressed with scaled average expression level at 1.3) (*Figure 1E*, *Figure 1—source data 1*). *Csf1* expression was negligible in osteo cluster and osteoblasts (scaled average expression level at 0, *Figure 1E*, *Figure 1—source data 1*). These results indicate that the Adipoq-lineage progenitors produce substantially more *Csf1* than the osteoblast lineage cells.

We next asked whether the Adipoq-lineage progenitors exist in human bone marrow. We analyzed a recently published scRNAseq dataset based on human femur bone marrow (*Wang et al., 2021*), and identified a similar cell population (cluster 1, Adipoq-lineage progenitors) that expressed high *ADIPOQ*, bone marrow stromal marker genes *LEPR*, *CXCL12* and *KITLG*, *LPL* and *CEBPA*, but with nearly undetectable osteoblast lineage genes (*Figure 2A and B*, *Figure 2—source data 1*). This cell population was mostly highly enriched in *CSF1* expression across the clusters (*Figure 2C, D and E*). In an additional human scRNAseq dataset (*Li et al., 2022*), we also found a similar human bone marrow stromal cluster (cluster 5, Adipo) that simultaneously expresses high levels of *ADIPOQ* and *CSF1* (*Figure 2—figure supplement 1*). These results indicate that human bone marrow contains a cell population that is highly similar to the bone marrow Adipoq-lineage progenitors in mice with conserved robust expression of *CSF1*.

## M-CSF is expressed in the bone marrow Adipoq-lineage progenitors but nearly undetectable in mature adipocytes in bone marrow or peripheral adipose

As Adipoq is expressed in mature adipocytes (*Berry and Rodeheffer, 2013*), we wondered whether M-CSF is also produced by these cells. We sorted the Adipoq-lineage progenitors from the *Adipoq Cre-mTmG* mice, and isolated both white and brown lipid-laden mature adipocytes, as well as the lipid-laden mature bone marrow adipocytes (BMAd). We found that the mRNA expression of *Csf1* in bone marrow Adipoq-lineage progenitor cells is 20–30 fold higher than that of mature adipocytes (*Figure 3A*, *Figure 3—figure supplement 1*). The *Csf1* expression in bone marrow Adipoq-lineage progenitor cells is also much higher than that of the stromal vascular fraction (SVF) cells in white adipose (*Figure 3A*). With this striking difference in mRNA expression, we further performed immunofluorescence staining of M-CSF on bone slices. We observed that the majority of bone marrow Adipoq-expressing progenitor cells express M-CSF (*Figure 3B*, 1865 cells out of 2001 cells counted, n=3 mice, 93.2%). In contrast, M-CSF expression was not detected in mature bone marrow adipocytes (Perilipin1+) (*Figure 3C*, 0 cells out of 115 cells counted, n=3 mice, 0%), indicating that mature bone marrow adipocytes are unlikely to be a significant source of M-CSF. Moreover, we performed western

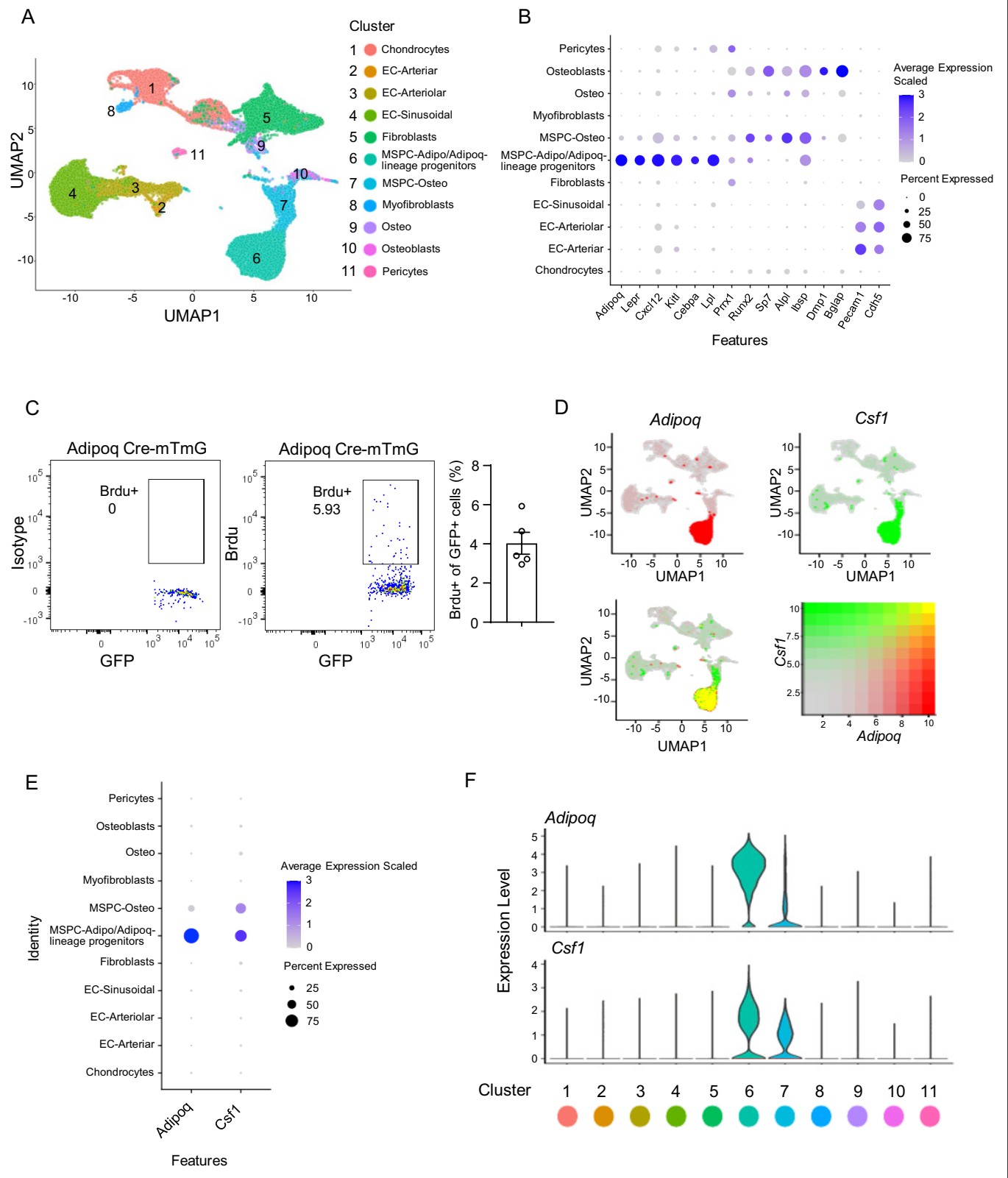

**Figure 1.** Integrated analysis of the bone marrow niche datasets of scRNAseq shows that Adipoq-lineage progenitors (Adipoq+ MSPCs) express high level of *Csf1*. (**A**) UMAP plot of the integrated analysis of the bone marrow niche datasets of scRNAseq based on **Dolgalev and Tikhonova, 2021**. EC, endothelial cell; MSPC: mesenchymal progenitor cell. (**B**) Dot plot of several typical marker gene expression for bone marrow stromal cells, adipocyte lineage, osteoblast lineage and endothelial cells across the listed scRNAseq clusters. Cell clusters are listed on the y-axis. Features are listed along the

*Figure 1 continued on next page*

Figure 1 continued

x-axis. Dot size reflects the percentage of cells in a cluster expressing each gene. Dot color reflects scaled average gene expression level as indicated by the legend. (**C**) Flowcytometry images and quantification of the bone marrow Adipoq-lineage progenitors (Adipoq+) incorporating BrdU in 10-week-old female Adipoq Cre-mTmG mice. n=5. (**D**) UMAP plots of the expression of *Adipoq* (upper left panel), *Csf1* (upper right panel) and the co-expression of these two genes (lower left panel) in bone marrow cells. The lower right panel shows a relative expression scale for each gene. (**E**) Dot plot of *Adipoq* and *Csf1* expression across the listed scRNA-seq clusters. Cell clusters are listed on y-axis. Features are listed along the x-axis. Dot size reflects the percentage of cells in a cluster expressing each gene. Dot color reflects the scaled average gene expression level as indicated by the legend. (**F**) Violin plots of the expression of *Adipoq* and *Csf1*.

The online version of this article includes the following source data for figure 1:

**Source data 1.** Integrated analysis of the bone marrow niche.

blot to analyze M-CSF protein expression in peripheral adipose tissue. As shown in *Figure 3D*, the stromal vascular fraction (SVF) cells in adipose, which contain multiple cell populations including adipogenic progenitors, express M-CSF. On the contrary, M-CSF was nearly undetectable in the peripheral mature adipocytes isolated from inguinal and epididymal white adipose tissue (WAT) (*Figure 3D*). These data collectively support that mature adipocytes are not a significant source of M-CSF as evidenced by nearly undetectable M-CSF expression compared to the Adipoq-lineage progenitors. These results identify Adipoq-lineage progenitors residing in bone marrow as a new cell type highly expressing M-CSF.

## Adipoq Cre-driven *Csf1* conditional knock out (Csf1$^{\Delta Adipoq}$) mice exhibit osteopetrosis

We next sought to investigate the contribution of the M-CSF produced by bone marrow Adipoq-lineage progenitors to bone development and homeostasis. We generated *Csf1* conditional knock out (KO) mice, in which *Csf1* is specifically deleted in Adipoq + cells by crossing *Csf1$^{flox/flox}$* mice with *Adipoq Cre* mice (*Csf1$^{f/f}$;AdipoqCre;* hereafter referred to as Csf1$^{\Delta Adipoq}$). Their littermates with a *Csf1$^{f/f}$* genotype were used as the controls. Compared to control mice, *Csf1* expression was reduced by approximately 75% in a total bone marrow stromal cell culture derived from the Csf1$^{\Delta Adipoq}$ mice (*Figure 4A*). Given that bone marrow Adipoq-lineage progenitors constitute only about 0.08% of bone marrow cells (*Figure 4—figure supplement 1*), these results, together with the data shown in *Figures 1–3*, support that bone marrow Adipoq-lineage progenitors are the major cellular source of M-CSF expression in the bone marrow. Furthermore, immunofluorescence staining of bone slices showed a drastic decrease in M-CSF protein expression in bone marrow Adipoq-lineage progenitor cells in Csf1$^{\Delta Adipoq}$ mice compared to the control mice (*Figure 4B*). Although SVF cells express Csf1, these cells do not express Adipoq (*Figure 3D*). Thus, it is unlikely that Csf1 expression in SVF is changed in Csf1$^{\Delta Adipoq}$ mice. Indeed, our results showed that *Csf1* expression in SVF was not altered in Csf1$^{\Delta Adipoq}$ mice (*Figure 4—figure supplement 2*). These results clearly demonstrate that adipoq-cre does not target SVF cells. We also examined the expression of a group of cytokines that often regulate macrophage function and osteoclastogenesis in bone marrow, including *Il34, Il1b, Il6, Il10, Csf2, Tnf, Cxcl12*, and found that the deficiency of *Csf1* in Csf1$^{\Delta Adipoq}$ mice did not affect the expression of these genes (*Figure 4—figure supplement 3*). Csf1$^{\Delta Adipoq}$ mice did not display abnormalities in gross appearance, body weight, tooth eruption and long bone length (*Figure 4C and D*, *Figure 4—figure supplement 4*). In contrast, microcomputed tomographic (μCT) analyses showed that Csf1$^{\Delta Adipoq}$ mice exhibited a marked osteopetrotic phenotype, as indicated by a twofold increase in trabecular bone mass and marked increases in bone mineral density (BMD), connectivity density (Conn-Dens.), trabecular bone number and a decrease in trabecular bone spacing compared to the littermate control mice (*Figure 4E and F*). Cortical bone appeared normal in Csf1$^{\Delta Adipoq}$ mice (*Figure 4G*). In addition, heterozygous *Csf1* conditional knockout mice (*Csf1$^{f/+}$;Adipoq Cre*) did not show an abnormal bone phenotype compared to control mice (*Csf1$^{f/f}$*) (*Figure 4—figure supplement 5*). There are no differences in vertebral BMD or bone mass between the control and Csf1$^{\Delta Adipoq}$ mice (*Figure 4—figure supplement 6*). Given that peripheral mature adipocytes (Adipoq + cells in peripheral adipose tissue) and mature bone marrow adipocytes (Adipoq +lipid-laden cells in bone marrow) express negligible levels of M-CSF, these data indicate that *Csf1* in bone marrow Adipoq-lineage progenitors plays a key role in the bone mass maintenance of long bones.

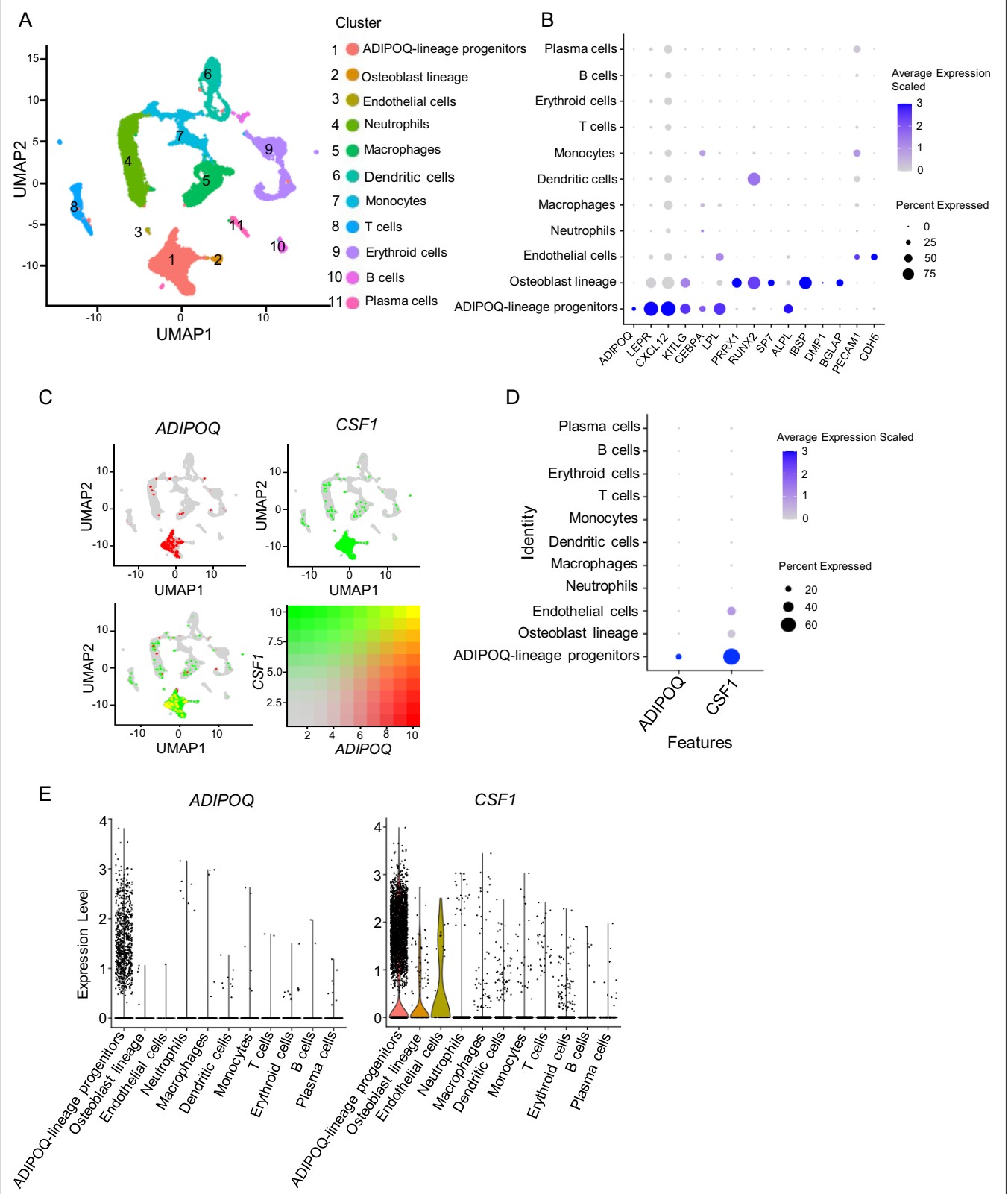

**Figure 2.** scRNAseq analysis of human bone marrow unveils the existence of ADIPOQ-lineage progenitors highly expressing *CSF1*. (**A**) UMAP plot analysis of the human bone marrow datasets of scRNAseq based on *Wang et al., 2021*. (**B**) Dot plot of several typical marker gene expression for bone marrow stromal cells, adipocyte lineage, osteoblast lineage and endothelial cells across the listed scRNA-seq clusters. Cell clusters are listed on the y-axis. Features are listed along the x-axis. Dot size reflects the percentage of cells in a cluster expressing each gene. Dot color reflects scaled average

*Figure 2 continued on next page*

*Figure 2 continued*

gene expression level as indicated by the legend. (**C**) UMAP plots of the expression of *ADIPOQ* (upper left panel), *CSF1* (upper right panel) and the co-expression of these two genes (lower left panel) in bone marrow cells. The lower right panel shows a relative expression scale for each gene. (**D**) Dot plot of *ADIPOQ* and *CSF1* expression across the listed scRNAseq clusters. Cell clusters are listed on y-axis. Features are listed along the x-axis. Dot size reflects the percentage of cells in a cluster expressing each gene. Dot color reflects the scaled average gene expression level as indicated by the legend. (**E**) Violin plots of the expression of *ADIPOQ* and *CSF1*.

The online version of this article includes the following source data and figure supplement(s) for figure 2:

**Source data 1.** scRNAseq analysis of human bone marrow.

**Source data 2.** Markers for cell type annotation (*Baryawno et al., 2019*; *Baccin et al., 2020*; *Dolgalev and Tikhonova, 2021*; *Wang et al., 2021*; *Komori et al., 1997*; *Nakashima et al., 2002*; *Diegel et al., 2020*; *Wang et al., 2018*; *Meijer et al., 2012*; *Bajpai et al., 2018*; *Fuentes-Duculan et al., 2010*; *Ippolito et al., 2014*; *Esashi et al., 2008*; *Honda et al., 2005*; *Kapellos et al., 2019*; *Bian et al., 2020*; *Mathewson et al., 2021*; *Shi et al., 2019*; *Ledergor et al., 2018*) in *Figure 2*.

**Figure supplement 1.** Heatmap of *ADIPOQ* and *CSF1* gene expression by LogFC (fold changes) values across the cell clusters based on human bone marrow scRNAseq dataset (*Li et al., 2022*).

## Bone marrow macrophages and osteoclasts are suppressed in Csf1$^{\Delta Adipoq}$ mice

The osteopetrotic phenotype in Csf1$^{\Delta Adipoq}$ mice implicated a defect in osteoclast function. Indeed, *Csf1* deficiency in Adipoq+ cells impaired osteoclast formation in vivo evidenced by reduced osteoclast surface area and lower osteoclast numbers (*Figure 5A*). The TRAP level in serum was significantly lower in Csf1$^{\Delta Adipoq}$ mice than that in control mice (*Figure 5B*). On the other hand, mineral apposition rate (MAR) and bone formation rate (BFR) were not affected in Csf1$^{\Delta Adipoq}$ mice, indicating normal osteoblastic function in these mice (*Figure 5—figure supplement 1*). Since osteoclasts are derived from the myeloid macrophage lineage, we examined bone marrow macrophage populations. CD11b+Ly6 C$^{hi}$ monocytes were similar between control and Csf1$^{\Delta Adipoq}$ mice. CD11b+F4/80+macrophages were reduced by almost half in Csf1$^{\Delta Adipoq}$ mice (*Figure 5C*). This decrease in macrophages reflects the effects of M-CSF deficiency, as M-CSF is critical for macrophage development. We further stimulated bone-marrow-derived macrophages (BMMs) with LPS and found that the inflammatory response of BMMs, indicated by inflammatory gene induction (*Figure 5—figure supplement 2A*) and the activation of MAPK or NF-κB pathways in response to LPS (*Figure 5—figure supplement 2B*), was similar between the BMMs derived from the control and Csf1$^{\Delta Adipoq}$ mice. To further test the importance of M-CSF produced by bone marrow Adipoq-lineage progenitors for osteoclastogenesis, we cultured whole bone marrow ex vivo without exogenous M-CSF to test whether the Adipoq-lineage cell-produced M-CSF is sufficient to function together with RANKL to induce osteoclast differentiation. As shown in *Figure 5D*, RANKL can induce osteoclast differentiation in the control bone marrow cultures even without addition of M-CSF, but it failed to induce osteoclastogenesis in the Csf1$^{\Delta Adipoq}$ bone marrow culture. When recombinant M-CSF was added back to the bone marrow cultures, the osteoclast formation in Csf1$^{\Delta Adipoq}$ cell cultures was similar to that in the control cultures (*Figure 5E*). These results demonstrate that the M-CSF secreted by bone marrow resident Adipoq-lineage progenitors is critical for osteoclastogenesis.

## *Csf1* deficiency in bone marrow Adipoq-lineage progenitors does not affect macrophage development in peripheral adipose tissue and spleen

Besides bone marrow, peripheral adipose tissue contains a large amount of Adipoq +mature adipocytes. However, M-CSF expression is undetectable in these cells (*Figure 3A and D*). In addition, some organs, such as spleen, have many tissue macrophages but few Adipoq + cells. We then asked whether *Csf1* expressed by bone marrow Adipoq-lineage progenitors affects macrophages outside of the bone marrow, such as in peripheral adipose tissue or the spleen. The gross appearance and weight of the spleen and peripheral adipose tissue, including the inguinal and epididymal adipose depots, were normal in Csf1$^{\Delta Adipoq}$ mice (*Figure 6A and B*). There was no difference in CD11b+Ly6 C$^{hi}$ monocytes in either the spleen, inguinal or epididymal adipose tissue between control and Csf1$^{\Delta Adipoq}$ mice, and the amount of CD11b+F4/80+macrophages was unchanged in these tissues (*Figure 6C*). Thus, in

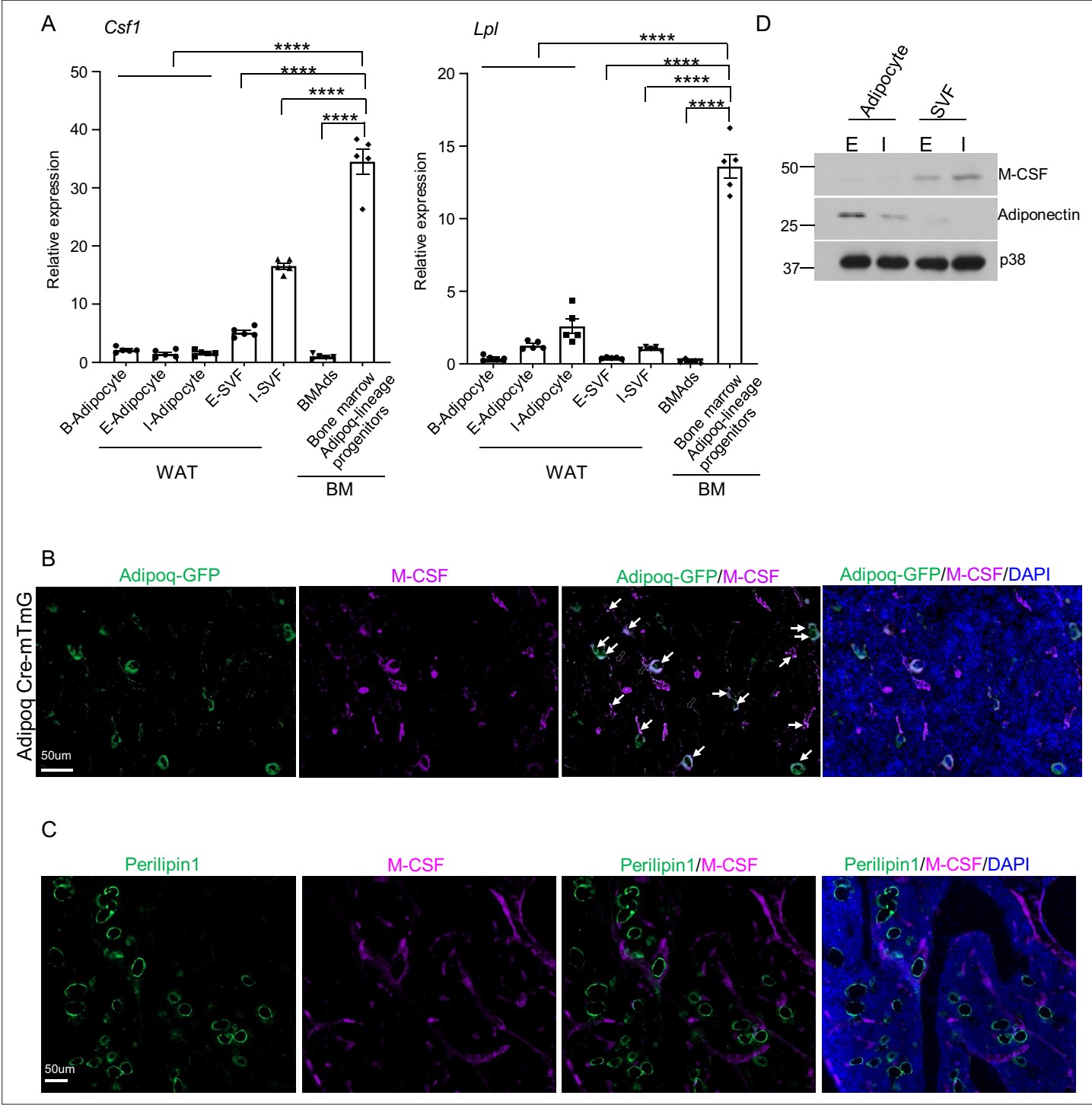

**Figure 3.** M-CSF is mainly produced by the bone marrow Adipoq-lineage progenitors, but not by mature adipocytes in peripheral adipose or in bone marrow. (**A**) qPCR analysis of *Csf1* and *Lpl* expression in bone marrow Adipoq-linage progenitors that were sorted from the bone marrow of Adipoq Cre-mTmG reporter mice, mature bone marrow adipocytes (BMAd), mature peripheral white and brown adipocytes and white stromal vascular fraction (SVF). E-adipocyte: mature adipocytes isolated from the epididymal white adipose tissue. I-adipocyte: mature adipocytes isolated from inguinal white adipose tissue. B-adipocyte: Brown adipocytes. n=5 for E-, I- and B-adipocytes from 12-week-old male mice. Five replicates, each with a pooled sample from 12-week-old male mice for BMAd (6–7 mice) and bone marrow Adipoq-lineage progenitors (3–4 mice). Error bars: Data are mean ± SD. ****p<0.0001 by one-way ANOVA analysis followed by post hoc Bonferroni's correction for multiple comparisons. (**B**) Immunofluorescence staining of M-CSF (purple) on femur bone slices from 12-week-old male Adipoq Cre-mTmG reporter mice. DAPI: blue. Arrows: co-localization of M-CSF and Adipoq-GFP in GFP +Adipoq-lineage progenitors. n=3 mice. (**C**) Immunofluorescence staining of M-CSF (purple) and Perilipin1 (green, mature adipocyte marker) on

*Figure 3 continued on next page*

*Figure 3 continued*

femur bone slices from 12-week-old male mice. n=3. (**D**) Immunoblot analysis of M-CSF and Adiponectin expression in mature adipocytes and stromal vascular fraction (SVF) in peripheral adipose. (**E**) epididymal white adipose tissue. (**I**) the inguinal white adipose tissue. p38 was used as a loading control.

The online version of this article includes the following source data and figure supplement(s) for figure 3:

**Source data 1.** M-CSF is mainly produced by the bone marrow Adipoq-lineage progenitors, but not by mature adipocytes in peripheral adipose or in bone marrow.

**Figure supplement 1.** qPCR analysis of *Csf1* expression relative to the geometric mean (*Vandesompele et al., 2002*) of *Gapdh* and *Actb* in bone marrow Adipoq-linage progenitors that were sorted from the bone marrow of Adipoq Cre-mTmG reporter mice, mature bone marrow adipocytes (BMAd), mature peripheral white and brown adipocytes.

**Figure supplement 1—source data 1.** qPCR analysis of *Csf1* expression relative to the geometric mean (*Vandesompele et al., 2002*) of *Gapdh* and *Actb* in bone marrow Adipoq-linage progenitors that were sorted from the bone marrow of Adipoq Cre-mTmG reporter mice, mature bone marrow adipocytes (BMAd), mature peripheral white and brown adipocytes.

contrast to the effects on bone marrow macrophages, *Csf1* deficiency in bone marrow Adipoq-lineage progenitors does not influence macrophage development in peripheral adipose tissue or the spleen.

## Lack of *Csf1* in bone marrow Adipoq-lineage progenitors alleviates estrogen-deficiency induced osteoporosis

We next investigated the significance of M-CSF secreted by Adipoq+ cells in pathological bone loss. We developed the ovariectomy (OVX) model in Csf1$^{\Delta\Delta Adipoq}$ mice to study the contribution of Adipoq+ cell-produced M-CSF to the estrogen-deficiency induced osteoporosis, which mimics postmenopausal bone loss. Uterine weight was measured six weeks after surgery to assess the success of the OVX. Uterine weights were decreased approximately 75% in OVX groups compared with the sham group (*Figure 7A*), indicating comparable and effective estrogen depletion in both control and Csf1$^{\Delta\Delta Adipoq}$ mice. OVX did not alter *Csf1* expression in the bone marrow (*Figure 7—figure supplement 1*). μCT analysis showed that OVX significantly reduced bone mass indicated by a decrease in BV/TV, trabecular number, trabecular thickness, Conn-Dens. and an increase of trabecular spacing compared with the sham group in the control mice (*Figure 7B and C*). Although OVX also decreased the bone mass of Csf1$^{\Delta\Delta Adipoq}$ mice, the extent of bone loss was less than that of control mice (*Figure 7B and C*). Significant changes were identified between the control and Csf1$^{\Delta\Delta Adipoq}$ mice in several μCT parameters. For example, a decrease in trabecular BV/TV after OVX: 35.1% in the control vs 20.9% in Csf1$^{\Delta\Delta Adipoq}$ mice; a decrease in Tb. N after OVX:11.34% in the control vs 7.97% in Csf1$^{\Delta\Delta Adipoq}$ mice; a decrease in Conn-Dens after OVX: 39.7% in the control vs 14.56% in Csf1$^{\Delta\Delta Adipoq}$ mice; an increase in Tb. Sp after OVX: 12.51% in the control vs 1.97% in Csf1$^{\Delta\Delta Adipoq}$ mice. Furthermore, the bone mass was markedly higher in Csf1$^{\Delta\Delta Adipoq}$ mice than control mice after OVX (*Figure 7B and C*, column 4 vs 2), indicating a significant role for the M-CSF secreted by Adipoq + cells in estrogen-deficiency induced bone loss. OVX is an osteoclastogenic stimulus. Thus, osteoclast formation as indicated by osteoclast surface and numbers was significantly enhanced by OVX in control mice (*Figure 7D*). Bone histomorphometric analysis further showed significantly greater osteoclast numbers and surface area in control mice with OVX than Csf1$^{\Delta\Delta Adipoq}$ mice with OVX (*Figure 7D*). Despite an increase in osteoclast surface in Csf1$^{\Delta\Delta Adipoq}$ mice after OVX, osteoclast numbers were not significantly influenced by OVX (*Figure 7D*). As bone marrow Adipoq-lineage progenitors are a key cell population expressing M-CSF, these results indicate that the absence of M-CSF in these cells appears to enable osteoclast linage cells to be resistant to environmental osteoclastogenic stimuli, such as OVX, thereby mitigating pathologic bone loss. These data demonstrate an important role for M-CSF produced by bone marrow Adipoq-lineage progenitors in pathologic osteoclastogenesis and bone loss.

## Discussion

M-CSF is indispensable for myeloid lineage development and the differentiation and function of osteoclasts, thus playing a crucial role in the physiology and pathology of the immune and skeletal systems. In this study, we identified bone marrow Adipoq-lineage progenitors as a new cellular source of M-CSF expression, which contributes substantially to the bone marrow macrophage development, physiological bone mass maintenance and pathological bone destruction via controlling osteoclast

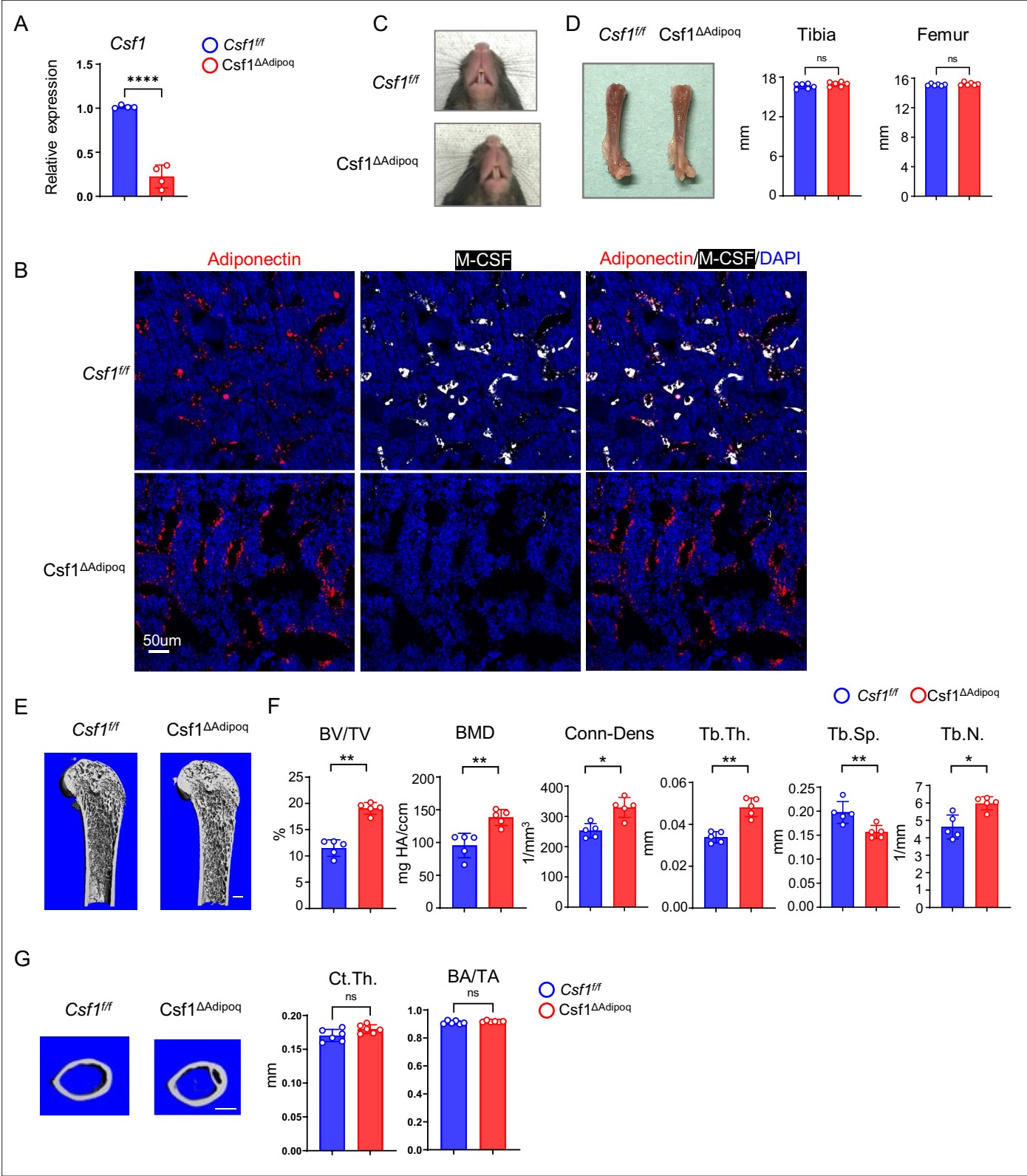

**Figure 4.** Csf1 deficiency in Csf1^ΔAdipoq mice increases bone mass. (**A**) *Csf1* expression in BMSCs derived from *Csf^f/f* and Csf1^ΔAdipoq (n = 4/group). (**B**) Representative images of immunostaining of Adiponectin (red) and M-CSF (white) on femur bone slices from 12-week-old male *Csf^f/f* and Csf1^ΔAdipoq mice. DAPI: blue. n=3/group. (**C**) Gross appearance of the incisors from *Csf^f/f* and Csf1^ΔAdipoq mice. (**D**) Gross appearance of the femur (left panel), and the lengths of femur and tibia from *Csf^f/f* and Csf1^ΔAdipoq mice (right panels) (n = 6/group). (**E**) μCT images and (**F**) bone morphometric analysis of

*Figure 4 continued on next page*

*Figure 4 continued*

trabecular bone of the distal femurs isolated from 12-week-old male *Csf* ^f/f^ and Csf1^ΔΔAdipoq^ mice (n = 5/group). (**G**) μCT images and bone morphometric analysis of cortical bone of the mid-shaft femurs isolated from 12-week-old male *Csf* ^f/f^ and Csf1^ΔΔAdipoq^ mice (n = 5/group). BM, bone marrow; BMSC, bone marrow stromal cell; BV/TV, bone volume per tissue volume; BMD, bone mineral density; Conn-Dens, connectivity density; Tb.Th, trabecular thickness; Tb.Sp, trabecular separation; Tb.N, trabecular number. Ct.Th, cortical bone thickness; BA/TA: Bone area/Tissue area. A, **D, F, G** *p<0.05; **p<0.01; ***p<0.001; ****p<0.0001; ns: not statistically significant by two tailed unpaired Student's t test. Error bars: Data are mean ± SD. Scale bars: B, 50 μm; **E**, 500 μm; **G**, 500 μm.

The online version of this article includes the following source data and figure supplement(s) for figure 4:

**Source data 1.** Csf1 deficiency in Csf1^ΔΔAdipoq^ mice increases bone mass.

**Figure supplement 1.** The flowcytometry images (left) and the percentage (right) of Adipoq-lineage progenitors (GFP+) in bone marrow (without red blood cells) from 12-week-old male mice.

**Figure supplement 1—source data 1.** The percentage of Adipoq-lineage progenitors (GFP+) in bone marrow (without red blood cells).

**Figure supplement 2.** qPCR analysis of *Csf1* expression in SVF from 12-week-old male *Csf* ^f/f^ and Csf1^ΔΔAdipoq^ mice (n = 3 for *Csf* ^f/f^ and n=2 for Csf1^ΔΔAdipoq^ mice).

**Figure supplement 2—source data 1.** qPCR analysis of *Csf1* expression in SVF.

**Figure supplement 3.** qPCR analysis of cytokines expressed in bone barrow from 12-week-old male *Csf* ^f/f^ and Csf1^ΔΔAdipoq^ mice (n = 3/group; For *Il34*, *Csf* ^f/f^ n=4, Csf1^ΔΔAdipoq^ n=6).

**Figure supplement 3—source data 1.** qPCR analysis of cytokines expressed in bone barrow.

**Figure supplement 4.** Gross appearance and body weight of 12-week-old male *Csf* ^f/f^ and Csf1^ΔΔAdipoq^ mice (n = 6/group).

**Figure supplement 4—source data 1.** Body weight of 12-week-old male *Csf* ^f/f^ and Csf1^ΔΔAdipoq^ mice.

**Figure supplement 5.** Bone morphometric analysis of trabecular bone of the distal femurs isolated from 12-week-old male *Csf* ^f/f^ and Csf1^f/+ΔAdipoq^ mice (n = 5/group).

**Figure supplement 5—source data 1.** Bone morphometric analysis of trabecular bone of distal femurs.

**Figure supplement 6.** μCT images (**A**) and bone morphometric analysis (**B**) of lumbar vertebrae (L5) isolated from 12-week-old male *Csf* ^f/f^ and Csf1^ΔΔAdipoq^ mice (n = 5/group).

**Figure supplement 6—source data 1.** Bone morphometric analysis of lumbar vertebrae (L5).

formation. Bone marrow Adipoq-lineage progenitors also produce RANKL (*Yu et al., 2021*; *Hu et al., 2021*), an essential cytokine to induce osteoclast differentiation. These findings collectively highlight the significance of bone marrow Adipoq-lineage progenitors in the regulation of osteoclastogenesis and bone metabolism, as well as the potential translational implications of appropriately targeting this cell population in treating pathologic bone loss.

The osteopetrotic phenotype in long bones is similar between Adipoq Cre-driven *Csf1* (our results) and RANKL conditional KO mice (*Yu et al., 2021*). Interestingly, Adipoq Cre-driven RANKL conditional KO mice also exhibit osteopetrosis in lumbar bone (*Yu et al., 2021*), while Csf1^ΔΔAdipoq^ mice do not. The mechanisms underlying this difference are unclear. A possibility is that cells other than Adipoq-expressing cells in lumbar presumably produce sufficient M-CSF or IL34 that could compensate the loss of M-CSF by Adipoq-lineage progenitors. This difference also implicates the presence of distinct cellular compartments and microenvironments between long bones and vertebral bones.

Adipoq is not only highly expressed in the bone marrow Adipoq-lineage progenitors but also in lipid-laden adipocytes in fat tissues (*Berry and Rodeheffer, 2013*), such as peripheral inguinal and epididymal adipose tissue. However, unlike the wide expression of Adipoq in lipid-laden adipocytes to Adipoq-lineage progenitors in bone marrow, Adipoq is expressed in mature lipid-laden adipocytes but not their progenitors in peripheral adipose tissue (*Berry and Rodeheffer, 2013*). Additionally, unlike bone marrow Adipoq+ progenitors, these peripheral Adipoq+ cells nearly do not produce M-CSF. The cells producing RANKL usually appear to simultaneously express M-CSF, such as osteocytes and bone marrow Adipoq-lineage progenitors (our findings and *Werner et al., 2020*; *Zhong et al., 2020*; *Yu et al., 2021*; *Hu et al., 2021*). This is not the case for peripheral Adipoq+ adipocytes, which express neither RANKL (*Fan et al., 2017*) nor M-CSF (our findings). The unique expression of M-CSF and RANKL by bone marrow Adipoq-lineage progenitors is a feature that distinguishes this cell population from other adipose depots.

Bone marrow macrophages are reduced by approximately half in the Csf1^ΔΔAdipoq^ mice, but macrophages in other tissues/organs are not significantly affected. These results indicate a strong local

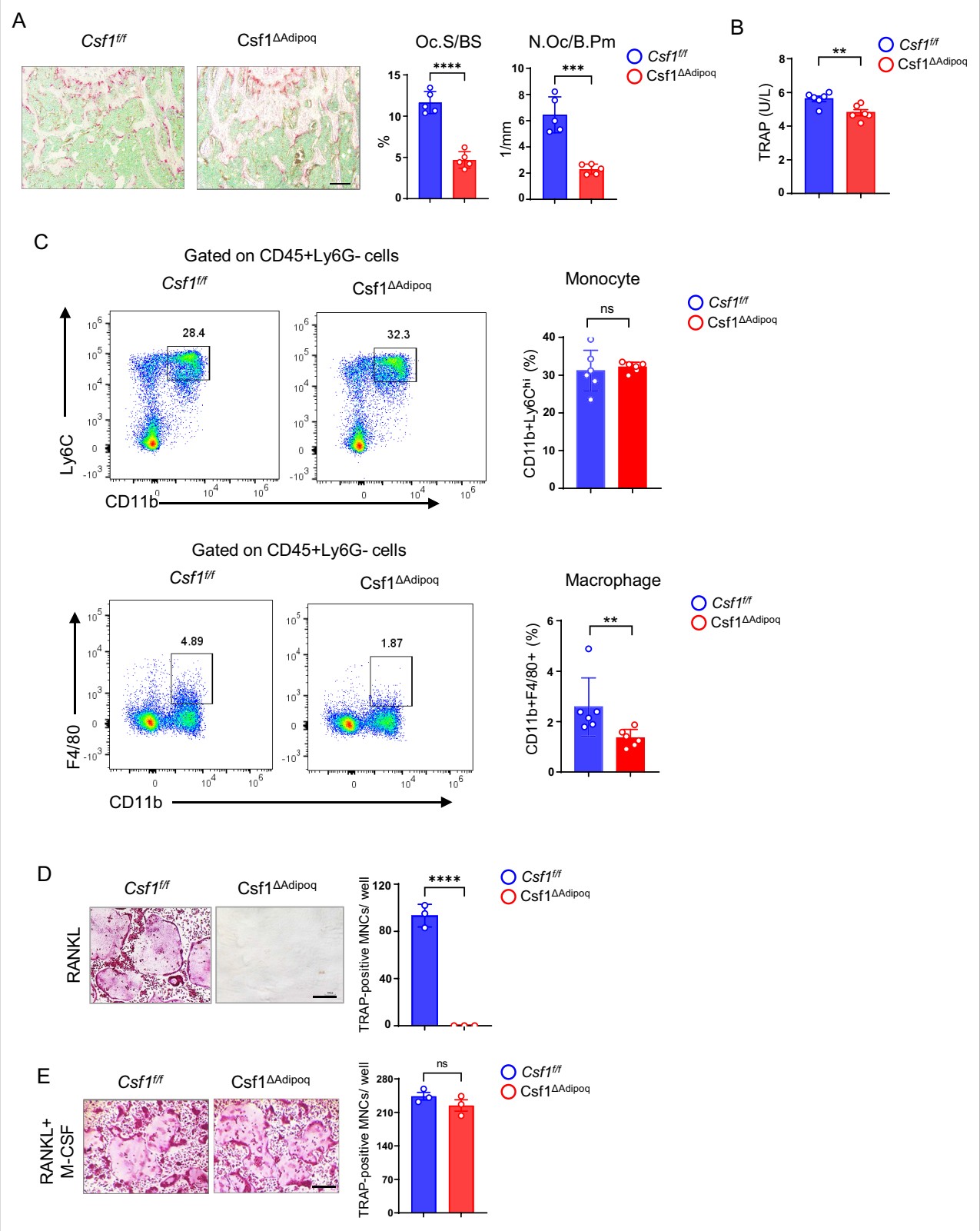

**Figure 5.** Csf1 deficiency in Csf1ΔΔAdipoq mice suppresses the populations of bone marrow macrophages and osteoclasts. (**A**) TRAP staining and histomorphometric analysis of histological sections obtained from the metaphysis region of distal femurs of 12-week-old male *Csf* f/f and Csf1ΔΔAdipoq mice (n = 5/group). (**B**) ELISA analysis of serum TRAP levels in 12-week-old male *Csf* f/f and Csf1ΔΔAdipoq mice (n = 6/group). (**C**) Flowcytometry image (left) and quantification (right) of monocytes and macrophages in bone marrow. n=6/group. (**D, E**) Osteoclast differentiation directly from the cultures of

*Figure 5 continued on next page*

*Figure 5 continued*

bone marrows harvested from *Csf*^f/f and Csf1^ΔAdipoq mice stimulated with RANKL (40 ng/ml) but without recombinant M-CSF for ten days (**D**) or with both RANKL and recombinant M-CSF (20 ng/ml) for five days (**E**). TRAP staining (left panel) was performed and the area of TRAP-positive MNCs (≥3 nuclei/cell) per well was calculated (right panel). TRAP-positive cells appear red in the photographs. (n = 3/group). Oc.S/BS, osteoclast surface per bone surface; N.Oc/B.Pm, number of osteoclasts per bone perimeter. (**A, B**), **C, D, E** **p<0.01; ***p<0.001; ****p<0.0001; ns: not statistically significant by two tailed unpaired Student's t test. Error bars: Data are mean ± SD. Scale bars: **A**, 100 μm; **D, E**, 200 μm.

The online version of this article includes the following source data and figure supplement(s) for figure 5:

**Source data 1.** Csf1 deficiency in Csf1^ΔAdipoq mice suppresses the populations of bone marrow macrophages and osteoclasts.

**Figure supplement 1.** Osteoblastic function in Csf1^ΔAdipoq mice is normal.

**Figure supplement 1—source data 1.** Dynamic histomorphometric analysis of trabecular bones (B) and cortical bones (C) of femurs.

**Figure supplement 2.** Inflammatory response to LPS is not altered in Csf1^ΔAdipoq BMMs.

**Figure supplement 2—source data 1.** (A) qPCR analysis of inflammatory gene expression and (B) Immunoblot analysis of the activation of signaling pathways in response to LPS stimulation (10 ng/ml) in BMMs.

regulation of macrophage development by bone marrow Adipoq-lineage progenitors via M-CSF. It was thought that M-CSF secreted by cells in certain tissues could affect M-CSF levels in other tissues/organs via circulation. However, research using *Csf1^op/ Csf1^op* mice with the restoration of circulating M-CSF shows that some tissues/organs are dependent much more on local than circulating M-CSF, such as bone marrow (*Cecchini et al., 1994*). Our results support this finding and further demonstrate that Adipoq-lineage progenitors are a crucial cellular source of M-CSF that controls macrophage and osteoclast populations in bone marrow. It is unclear but interesting why macrophage development

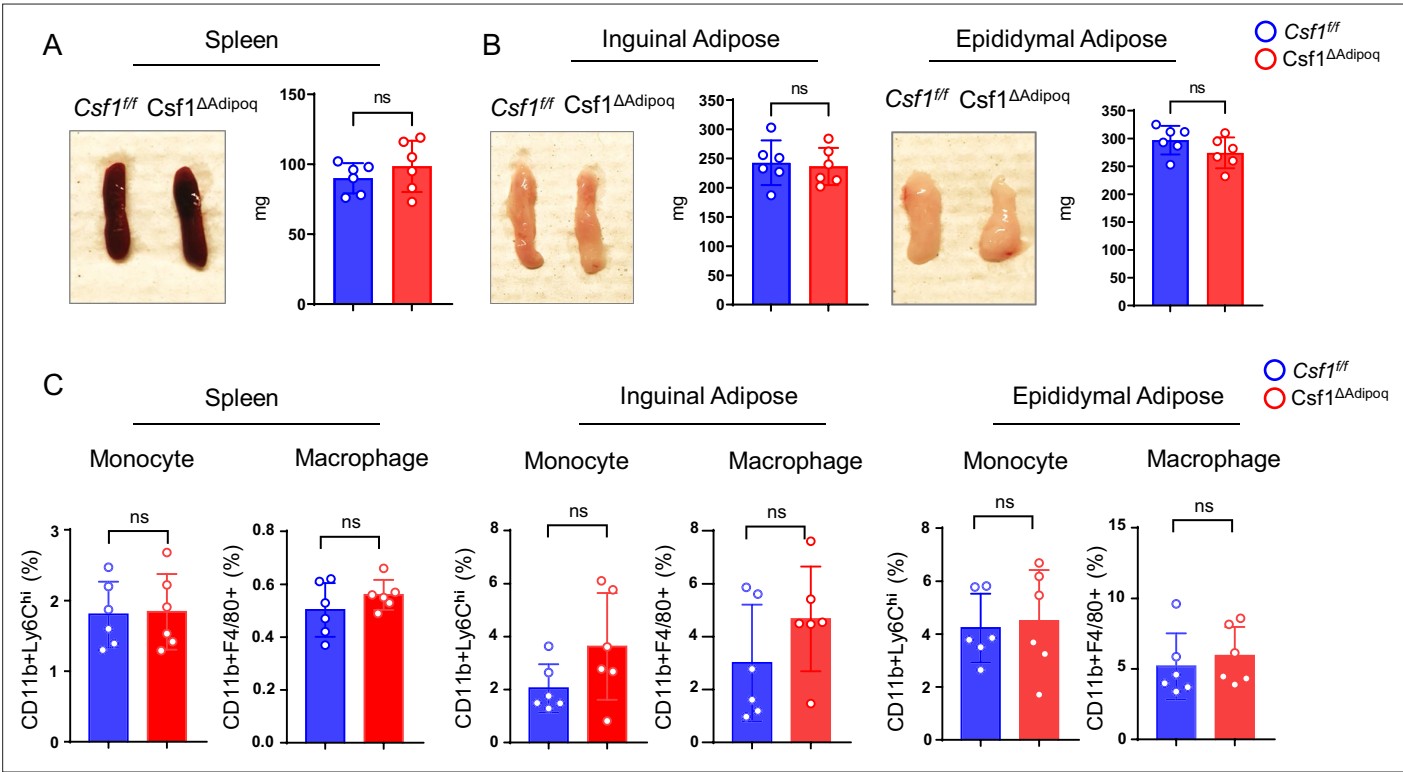

**Figure 6.** Csf1 deficiency in Csf1^ΔAdipoq mice does not affect monocyte and macrophage populations in spleen and peripheral adiposes. (**A**) Gross appearance (left panel) and weight (right panel) of the spleen from *Csf*^f/f and Csf1^ΔAdipoq mice (n = 6/group). (**B**) Gross appearance and weight of the inguinal (left panels) and the epididymal (right panels) adipose from *Csf*^f/f and Csf1^ΔAdipoq mice (n = 6/group). (**C**) Flowcytometry quantification of monocytes and macrophages (gated on CD45 +Ly6G- cells) in the indicated tissues. n=6/group. (**A, B, C**) ns: not statistically significant by two tailed unpaired Student's t test. Error bars: Data are mean ± SD.

The online version of this article includes the following source data for figure 6:

**Source data 1.** Csf1 deficiency in Csf1^ΔAdipoq mice does not affect monocyte and macrophage populations in spleen and peripheral adiposes.

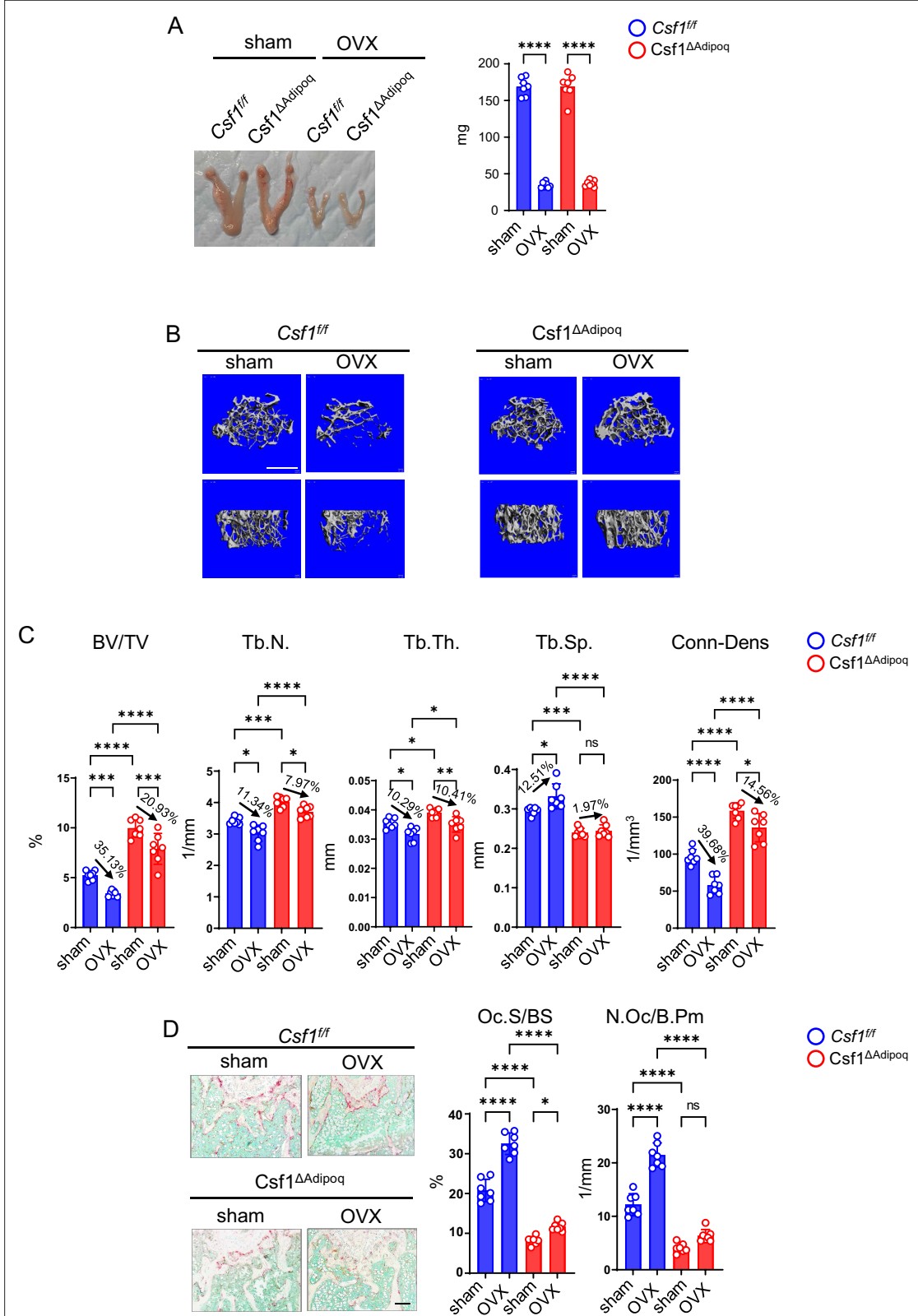

**Figure 7.** *Csf1* deficiency in Csf1ΔAdipoq mice protects bone in OVX model. 12-week-old female *Csf* f/f and Csf1ΔAdipoq mice were subjected to OVX or sham surgery and analyzed 6 weeks after surgery. (**A**) Gross appearance (left panel) and weight (right panel) of uterus, (**B**) μCT images, and (**C**) bone morphometric analysis of trabecular bone of the distal femurs isolated from the *Csf* f/f and Csf1ΔAdipoq mice with sham or OVX surgery (n = 7/group). (**D**) TRAP staining (left panels) and histomorphometric analysis (right panels) of histological sections obtained from the metaphysis region of distal

*Figure 7 continued on next page*

Figure 7 continued

femurs isolated from the indicated mice (n = 7/group). BV/TV, bone volume per tissue volume; Tb.N, trabecular number; Tb.Th, trabecular thickness; Tb.Sp, trabecular separation; Conn-Dens., connectivity density; Ct.Th, cortical thickness; Oc.S/BS, osteoclast surface per bone surface; N.Oc/B.Pm, number of osteoclasts per bone perimeter. Data are mean ± SD. *p<0.05; **p<0.01; ***p<0.001; ****p<0.0001; n.s., not statistically significant by two-way ANOVA analysis followed by post hoc Bonferroni's correction for multiple comparisons. Scale bars: **B**, 500 µm; **D**, 100 µm.

The online version of this article includes the following source data and figure supplement(s) for figure 7:

**Source data 1.** *Csf1* deficiency in Csf1$^{\Delta\Delta Adipoq}$ mice protects bone in OVX model.

**Figure supplement 1.** qPCR analysis of *Csf1* expression in the bone marrow from the Sham and OVX mice.

**Figure supplement 1—source data 1.** qPCR analysis of *Csf1* expression in the bone marrow from the Sham and OVX mice.

and osteoclastogenesis in bone marrow is more dependent on local M-CSF. This is presumably a unique mechanism by which bone marrow microenvironment decouples osteoclastogenesis and marrow monocytic development from systemic regulation of tissue-resident macrophages, allowing for marrow homeostasis.

The bone marrow Adipoq-lineage progenitors only constitute approximately 0.08% of bone marrow cells. However, the deficiency of M-CSF produced by this small Adipoq+ cell population leads to around a 50% increase in trabecular bone mass and an over 50% reduction of bone marrow macrophages. These results together with the scRNAseq data in *Figure 1* demonstrate that Adipoq-lineage progenitor cells in bone marrow produce substantially more M-CSF than osteoblast lineage cells. Thus, Adipoq-lineage progenitors are a major cellular source of M-CSF for bone marrow. We also observed that the osteopetrotic phenotype in Csf1$^{\Delta\Delta Adipoq}$ mice is not as strong as *Csf1$^{op}$/ Csf1$^{op}$* mice. Literature show that M-CSF expression was found in cultures of osteoblastic cells, an osteocyte cell line and osteocytes (*Tanaka et al., 1993*; *Werner et al., 2020*; *Elford et al., 1987*; *Zhao et al., 2002*). Despite scRNAseq analysis showing that osteoblasts do not express *Csf1*, the MSPC-Osteo population does (*Figure 1*). The possibility of M-CSF expression by osteoblasts also exists, which needs further assessment in vivo in addition to scRNAseq approach. Collectively, the M-CSF produced by MSPC-Osteo cells, osteocytes and potentially osteoblasts could partially compensate for the loss of M-CSF from bone marrow Adipoq-lineage progenitor cells.

M-CSF functions through its receptor c-Fms to exert various biological activities. IL34 was identified later as an additional ligand for c-Fms. Although it also binds to other receptors aside from c-Fms, IL34 has, at least partially, functional redundancy with M-CSF (*Lin et al., 2019*; *Lelios et al., 2020*). Unlike M-CSF that is broadly and highly expressed by almost every tissue, IL34 is found highly expressed in brain and skin and plays an important role in the differentiation and maintenance of microglia and Langerhans cells (*Lin et al., 2019*; *Lelios et al., 2020*). The vascular endothelial cells in spleen also produce IL34 (*Nakamichi et al., 2012*). The results from scRNAseq datasets show that IL34 is expressed in bone marrow, largely by Adipoq-lineage progenitors but also moderately by pericytes and slightly by MSPC-Osteo cells (*Dolgalev and Tikhonova, 2021*). This indicates that Adipoq-lineage progenitors are highly likely a major cellular source of IL34 in bone marrow. Spleen has a reservoir of RANK+/CSF-1*R*+osteoclast precursors, which can home to bone marrow to differentiate to mature osteoclasts in response to RANKL, IL34 and/or M-CSF (*Nakamichi et al., 2012*). *Il34* expression in bone marrow was not affected in Csf1$^{\Delta\Delta Adipoq}$ mice. These mechanisms collectively may partially compensate for M-CSF function and alleviate the osteopetrotic and macrophage deficiency phenotype in Csf1$^{\Delta\Delta Adipoq}$ mice.

Recently, increasing attention has been paid to the question on whether Adipoq-lineage progenitors consist exclusively of adipogenic cells. In addition to Adipoq-lineage progenitors, Adipoq Cre also labels other clusters. However, the expression levels of *Adipoq* and frequency of Adipoq+ cells in other cell populations are relatively low. For example, the integrated scRNAseq dataset we analyzed shows that *Adipoq* is expressed at a low level (with scaled mean expression at 0.68, *Dolgalev and Tikhonova, 2021*) in a small proportion of MSPC-osteo cells (*Figure 1*), and small amounts (*Yu et al., 2021*; *Jeffery et al., 2022*; about 4%) of osteoblasts in 8 or 12-week-old mice are Adipoq-lineage. A recent report found that in 24-week-old mice, about 15–40% of osteoblasts are marked with Adipoq Cre (*Jeffery et al., 2022*). This raises a few important possibilities that will need to be distinguished in future work. One possibility is that the Adipoq-lineage cells (adipo-CAR cells/MALPs) have minor or latent osteogenic potential that may become more evident under specific conditions, such as in

older animals. However, balanced against this is the alternative that Adipoq-cre could primarily target a population of solely adipogenic adipo-CAR cells but that its specificity is imperfect, leading to progressive low levels of deletion in a separate population expressing very low levels of Adipoq, such as osteo-CAR cells. An additional possibility is that the Adipoq-lineage cells may themselves actually be further subdivided into multiple component cell types, including a major adipogenic and a separate minor osteogenic subpopulation. Ultimately, at the root of these issues is that Adipoq cre primarily defines one or possibly more lineages of cells rather than a cell type within those lineages. Therefore, application of further markers to fractionate the adipoq-lineage into its component cell types will be needed to resolve these possibilities, focusing on whether any potential osteogenic activity present can be fractionated away from the primary adipogenic activity present.

Of note, the *Adipoq* expression level and positive cell proportion are much higher in bone marrow Adipoq lineage progenitors than the levels seen in osteoblast lineage (*Figure 1*, *Figure 2*, *Zhong et al., 2020*; *Dolgalev and Tikhonova, 2021*; *Yu et al., 2021*) or endothelial cells in bone marrow (*Zhang et al., 2021*; *Emoto et al., 2022*). For example, the MSPC-Adipo cluster (Adipoq-lineage progenitors) has 6441 cells with the highest level (scaled mean expression level at 3.01 per *Dolgalev and Tikhonova, 2021* at Single Cell Portal) of *Adipoq* seen among bone marrow cells analyzed. In contrast, the MSPC-osteo cluster consists of 2247 cells with a very low *Adipoq* expression level (scaled mean expression level at 0.68 per *Dolgalev and Tikhonova, 2021* at Single Cell Portal). Taken together with both average expression level and cell numbers in each cluster, the relative overall contribution to *Adipoq* expression by MSPC-osteo vs the Adipoq-lineage progenitors is 7.8% ([2247x0.68]/[6441x3.01]). Therefore, the expression of *Adipoq* in MSPC-osteo cluster is marginal compared to that in the Adipoq-lineage progenitors. These data make Adipoq as an important marker to identify bone marrow Adipoq lineage progenitors. Overall, our work not only validates prior research identifying adipoq-lineage cells, identified as MALPs (*Zhong et al., 2020*; *Yu et al., 2021*), as a key osteoclast regulatory population, but also further extends the scope of their functions to encompass M-CSF production and regulation of macrophages.

In addition to its function in development and postnatal homeostasis, M-CSF is a key factor contributing to pathological bone destruction. In OVX induced osteoporosis model, we found that the extent of bone loss in Csf1$^{\Delta\Delta Adipoq}$ mice was less than that of the control mice, and the bone mass was significantly higher in Csf1$^{\Delta\Delta Adipoq}$ mice than control mice after OVX. These results indicate a significant role for the M-CSF produced by Adipoq + cells in estrogen-deficiency induced bone loss. Interestingly, however, no genotype by OVX interaction term was seen in two-way ANOVA, even though there is a significant difference in the post-OVX bone mass between genotypes. The difference in baseline bone mass pre-OVX between genotypes could presumably complicate and compromise the significance analysis of genotype by OVX interaction. Further study, for example, using inducible Adipoq-cre mice, is needed to compare whether Adipoq-lineage progenitors-derived M-CSF specifically regulates OVX responses or pathological bone loss induced by other stimuli, or is instead more involved in basal homeostasis. Broad blockade of M-CSF, however, is not clinically feasible because this may suppress macrophage survival and function in a variety of important tissues and organs. Our results show that lack of M-CSF in bone marrow Adipoq-lineage progenitors does not affect macrophage populations outside of the bone marrow. These findings suggest that targeting bone marrow Adipoq-lineage progenitors themselves or the mechanisms regulating their production of M-CSF could be a potential therapeutic strategy to prevent pathological bone destruction.

## Methods

**Key resources table**

| Reagent type (species) or resource | Designation | Source or reference | Identifiers | Additional information |
|---|---|---|---|---|
| Genetic reagent (*M. musculus*) | Csf1$^{flox/flox}$ | PMID:21958845 | RRID:MGI:5305712 | |
| Genetic reagent (*M. musculus*) | Rosa26$^{mT/mG}$ | PMID:17868096 | RRID:IMSR_JAX:007676 | |

*Continued on next page*

*Continued*

| Reagent type (species) or resource | Designation | Source or reference | Identifiers | Additional information |
|---|---|---|---|---|
| Genetic reagent (*M. musculus*) | Adipoq-Cre | PMID:21356515 | RRID:IMSR_JAX:028020 | |
| Antibody | Anti-mouse CD45-PerCP/Cyanine5.5 (Rat monoclonal) | BioLegend | Cat# 103132 RRID: AB_893340 | FACS (1:200) |
| Antibody | Anti-mouse Ly-6G- Brilliant Violet 711 (Rat monoclonal) | BioLegend | Cat# 127643 RRID:AB_2565971 | FACS (1:200) |
| Antibody | Anti-mouse/human CD11b- Alexa Fluor 700 (Rat monoclonal) | BioLegend | Cat# 101222 RRID:AB_493705 | FACS (1:200) |
| Antibody | Anti-mouse F4/80- APC/Cyanine7 (Rat monoclonal) | BioLegend | Cat# 123118 RRID:AB_893477 | FACS (1:200) |
| Antibody | Anti-mouse Ly-6C- Brilliant Violet 510 (Rat monoclonal) | BioLegend | Cat# 128033 RRID:AB_2562351 | FACS (1:200) |
| Antibody | Anti-mouse CD45- APC (Rat monoclonal) | BioLegend | Cat# 103112 RRID:AB_312977 | FACS (1:200) |
| Antibody | Anti-mouse CD31-APC (Rat monoclonal) | BioLegend | Cat# 102510 RRID:AB_312917 | FACS (1:200) |
| Antibody | Anti-mouse TER-119 /Erythroid Cells- APC (Rat monoclonal) | BioLegend | Cat# 116212 RRID:AB_313713 | FACS (1:200) |
| Antibody | Anti-BrdU-APC (Mouse monoclonal) | BioLegend | Cat# 364114 RRID:AB_2814315 | FACS (5 ug per test) |
| Antibody | Mouse IgG1, κ Isotype-APC (Mouse monoclonal) | BioLegend | Cat# 400119 RRID:AB_2888687 | FACS (5 ug per test) |
| Antibody | Anti-Adiponectin (Rabbit polyclonal) | Thermo Fisher Scientific | Cat# PA1-054 RRID:AB_325789 | IF(1:200) WB(1:1000) |
| Antibody | Anti-Mouse IgG- Alexa Fluor 647 (Goat polyclonal) | Thermo Fisher Scientific | Cat# A-21235 RRID:AB_2535804 | IF(1:2000) |
| Antibody | Anti-Rabbit IgG- Alexa Fluor 594 (Goat polyclonal) | Thermo Fisher Scientific | Cat# A-11012 RRID:AB_2534079 | IF(1:2000) |
| Antibody | Anti-M-CSF (Mouse monoclonal) | Santa Cruz Biotechnology | Cat# sc-365779 RRID:AB_10846852 | IF(1:200) WB(1:1000) |
| Antibody | Anti-Perilipin (Rabbit monoclonal) | Cell Signaling Technology | Cat# 9349 RRID:AB_10829911 | IF(1:100) |
| Antibody | Anti-P38alpha (Rabbit polyclonal) | Santa Cruz Biotechnology | Cat# sc-535 RRID:AB_632138 | WB(1:1000) |
| Antibody | Anti-Phospho-NF-κB p65 (Ser536) (Rabbit monoclonal) | Cell Signaling Technology | Cat# 3033 RRID:AB_331284 | WB(1:1000) |
| Antibody | Anti-p44/42 MAP kinase (Rabbit polyclonal) | Cell Signaling Technology | Cat# 9101 RRID:AB_331646 | WB(1:1000) |
| Antibody | Anti-Phospho-SAPK/JNK (Thr183/Tyr185) (Rabbit polyclonal) | Cell Signaling Technology | Cat# 9251 RRID:AB_331659 | WB(1:1000) |
| Commercial assay or kit | eBioscience BrdU Staining Buffer | Thermo Fisher Scientific | Cat#: 00-5525-00 | |
| Peptide, recombinant protein | Recombinant Human sRANK Ligand | PeproTech | Cat# 310–01 | 40 ng/mL |
| Peptide, recombinant protein | Murine M-CSF | PeproTech | 315–02 | 20 ng/ml |
| Chemical compound, drug | 5-Bromo-2'-deoxyuridine (Brdu) | Sigma Aldrich | Cat#: B5002 | 200 mg/Kg |

*Continued on next page*

*Continued*

| Reagent type (species) or resource | Designation | Source or reference | Identifiers | Additional information |
|---|---|---|---|---|
| Commercial assay or kit | LIVE/DEAD Fixable Blue Dead Cell Stain Kit, for UV excitation | Thermo Fisher Scientific | L23105 | 1:1000 |
| Commercial assay or kit | Mouse Tartrate Resistant Acid Phosphatase (TRAP) ELISA Kit | MyBioSource.com. | MBS1601167 | |
| Software, algorithm | Seurat | PMID:29608179 | RRID:SCR_016341 | https://satijalab.org/seurat/get_started.html |
| Software, algorithm | Graphpad Prism 8 | GraphPad Software | RRID:SCR_002798 | |
| Software, algorithm | Flowjo V10.7.1 | Flowjo | RRID:SCR_008520 | https://www.flowjo.com/solutions/flowjo |
| Software, algorithm | ZEN (blue edition) version 3.4 | ZEN (blue edition) | RRID:SCR_013672 | https://www.zeiss.com/microscopy/en/products/software/zeiss-zen.html |
| Other | DAPI stain | BD Biosciences | Cat# 564907, RRID:AB_2869624 | FACS (1 ug/ml) |

## Mice and analysis of bone phenotype

*Csf1*$^{flox/flox}$ mice have been described previously (*Harris et al., 2012*). We generated mice with Adipoq+ cell specific deletion of *Csf1* by crossing *Csf1*$^{flox/flox}$ mice with the mice with an Adipoq promoter-driven Cre transgene on the C57BL/6 background (Adipoq-Cre: The Jackson Laboratory, stock No: 028020). We also generated Adipoq Cre-mTmG reporter mice by crossing the mTmG mice (The Jackson Laboratory, stock No: 007676) with the Adipoq-Cre mice. Sex- and age-matched *Csf1*$^{flox/flox}$;*Adipoq-Cre* mice (referred to as Csf1$^{ΔAdipoq}$) and their littermates with *Csf1*$^{flox/flox}$ genotype as controls (referred to as *Csf1*$^{f/f}$) were used for experiments. *Csf1*$^{flox/flox}$ mice were a gift from Dr. Jean X. Jiang (UT Health San Antonio). Bilateral ovariectomy (OVX) or sham operation (Sham) was performed on 12-week-old female mice. The mice were sacrificed 6 weeks after surgery. Uterine atrophy was first confirmed and then bones were collected for µCT and histological analysis. The mice with the same genotype were randomly allocated to different treatments or procedures. All mouse experiments were approved (protocol: 2016–0001 and 0004) by Institutional Animal Care and Use Committee of the Hospital for Special Surgery and Weill Cornell Medical College.

µCT analysis was conducted to evaluate bone volume and 3D bone architecture using a Scanco µCT-35 scanner (SCANCO Medical). Mice femora were fixed in 10% buffered formalin and scanned at 6 µm resolution. Proximal femoral trabecular bone parameters or the cortical bone parameters obtained from the midshaft of femurs were analyzed using Scanco software according to the manufacturer's instructions and the American Society of Bone and Mineral Research (ASBMR) guidelines. Femur bones were decalcified and subjected to sectioning, TRAP staining and histological analysis. For dynamic histomorphometric measures of bone formation, calcein (25 mg/kg, Sigma) was injected into mice intraperitoneally at 7 and 2 days before sacrifice to obtain double labeling of newly formed bones. The non-decalcified femur bones were processed with 5% aqueous potassium hydroxide for 96 hr, dehydrated and embedded in paraffin (*Porter et al., 2017*). 5 µm thick sections were sliced using a microtome. The Osteomeasure software was used for bone histomorphometry using standard procedures according to the program's instruction.

## Single-cell RNAseq (scRNAseq) analysis

The data from the integrated analysis of the three scRNA-seq datasets of the non-hematopoietic compartment of the murine bone marrow were extracted from the Open Science Framework (https://osf.io/ne9vj) (*Dolgalev and Tikhonova, 2021*). We further extracted the clusters of 'Chondrocytes', 'EC-Arteriar', 'EC-Arteriolar', 'EC-Sinusoidal', 'Fibroblasts', 'MSPC-Adipo', 'MSPC-Osteo', 'Myofibroblasts', 'Osteo', 'Osteoblasts', and 'Pericytes' from the integrated data and analyzed the data using Seurat (*Stuart et al., 2019*). A human bone marrow scRNAseq dataset (*Wang et al., 2021*) was also extracted and re-analyzed using Seurat. UMAP (uniform manifold approximation and projection) was used for dimensionality reduction after PCAs were calculated for integrated datasets. We

visualized the simultaneous expression of two genes in a cell using FeaturePlot function in Seurat. The percentage of cells in a cluster expressing a certain gene and the expression level of each gene were visualized by DotPlot function in Seurat. The distribution of gene expression in each cluster was visualized using VlnPlot function in Seurat. R version 4.0.2 (2020-06-22) and Seurat 4.0.2 were used for *Figure 1A, D and F*. R version 4.1.2 (2021-11-01) and Seurat 4.1.1 were used for the left bioinformatics analysis in the study.

## Cell culture

For osteoclastogenesis directly from bone marrow cultures, bone marrow cells were seeded at a density of $3.125 \times 10^5$ /cm², and cultured in α-MEM medium with 10% FBS, glutamine (2.4 mM, Thermo Fisher Scientific) and Penicillin–Streptomycin (Thermo Fisher Scientific) for 2 days. The cells were then treated with RANKL (40 ng/ml) without M-CSF for 10 days, or with both RANKL (40 ng/ml) and M-CSF (20 ng/ml) for 5 days. Medium were changed every 2 days.

## Reverse transcription and real-time PCR

DNA-free RNA was obtained with the RNeasy MiniKit (Qiagen, Valencia, CA) with DNase treatment according to the manufacturer's instructions. For the RNA extraction from the pooled Adipoq-progenitors or BMAds, 13 µl Rnase-free H2O was used to elute RNA from the column in the last step. Then, 1 µg of total RNA or total amount of RNA extracted from the pooled Adipoq-progenitors or BMAds was reverse-transcribed with random hexamers and MMLV-Reverse Transcriptase (Thermo Fisher Scientific) according to the manufacturer's instructions. Real-time PCR was done in triplicate with the QuantStudio 5 Real-time PCR system and Fast SYBR Green Master Mix (Thermo Fisher Scientific) with 500 nM primers. mRNA amounts were normalized relative to glyceraldehyde-3-phosphate dehydrogenase (GAPDH) mRNA. The primers for real-time PCR were as follows: *Csf1*: 5′-AAAGACAA CACCCCCAATGC-3′ and 5′-AGGAGTCTCATGGAAAGTTCGG-3′; *Lpl:* 5′-TTCCAGCCAGGATGCA ACA-3′ and 5′-GGTCCACGTCTCCGAGTCC-3′; *Gapdh*: 5′-ATCAAGAAGGTGGTGAAGCA-3′ and 5′-AGACAACCTGGTCCTCAGTGT-3′; *Actb:* 5′-CTGACTGACTACCTCATGAAGATCCT-3′ and 5′-CTTA ATGTCACGCACGATTTCC-3′; *Il6:* 5′-TACCACTTCACAAGTCGGAGGC-3′ and 5′-CTGCAAGTGCAT CATCGTTGTTC-3′; *Il1b:* 5′-AGCTTCCTTGTGCAAGTGTCT-3′ and 5′-GACAGCCCAGGTCAAAGGTT -3′; *Tnf:* 5′-CCCTCACACTCAGATCATCTTCT-3′ and 5′-CTTTGAGATCCATGCCGTTG-3′; *Csf2:* 5′- CCAGCTCTGAATCCAGCTTCTC-3′ and 5′-TCTCTCGTTTGTCTTCCGCTGT-3′; *Il34:* 5′- ACTCAGAG TGGCCAACATCACAAG-3′ and 5′-ATTGAGACTCACCAAGACCCACAG-3′; *Cxcl12:* 5′-TGCATCAG TGACGGTAAACCA-3′ and 5′-TTCTTCAGCCGTGCAACAATC-3′; *Il10:* 5′-GGTTGCCAAGCCTTAT CGGA-3′ and 5′-GGGGAGAAATCGATGACAGC.

## Frozen sectioning and immunofluorescence staining

Freshly dissected mouse bones were fixed with 4% paraformaldehyde (PFA) (15,710 S, Electron Microscopy Sciences) for 4 hr at 4 °C. After samples were washed with PBS, decalcification was processed with 0.5 M EDTA for 5 days. Samples were incubated with infiltration solution (20% sucrose (S7903, Sigma) plus 2% polyvinylpyrrolidone (Sigma, 9003-39-8) in PBS) until they sank to the bottom of the tube. Embedding was performed with OCT (Sakura Finetek,4583) and samples were stored at −80 °C. The samples were sectioned at 10 µm in thickness using a Leica cryostat. Frozen sections were thawed at room temperature and rehydrated with PBS, permeabilized with 0.5% Triton X-100 in PBS for 15 min at room temperature, and blocked for 1 hr with 5% BSA in PBS (blocking buffer). Primary antibodies were freshly diluted in blocking buffer and were incubated with the slices overnight at 4 °C. After washing three times with PBS, secondary antibodies (1:2000 dilution with blocking buffer) were incubated for 1 hr at room temperature. Samples were then washed and mounted with anti-fade mounting solution with DAPI (Life technologies, P36941). Imaging was employed using a Zeiss Axioscan 7 Slide Scanner.

## Flow cytometry

Bone marrow cells were directly flushed out by quick centrifuge (from 0 to 9400 g, approximately 15 s at room temperature) after cutting both ends of long bones, then resuspended by PBS (Corning, 21040CV) and filtered through 70 µm cell strainer (FALCON,352350). The cells were spun down at 500 g for 5 min at 4 °C. Epididymal white adipose tissue (EWAT) and Inguinal WAT (IngWAT) were

isolated from mice and minced into small pieces by scissors and then digested by the digestion buffer (a-MEM [Gibco, 12561056] with 2 mg/ml collagenase II [Worthington, LS004176] and 2% BSA [Gemini Bio, 700–100 P]) for 40 min in a rotary incubator at 250 rpm at 37 °C. The cells were then filtered through 70 µm cell strainer and spun down at 500 g for 5 min at 4 °C. One third of spleen was mashed with the plunger end of a syringe through the 70 µm cell strainer. The splenic cells were then spun down at 500 g for 5 min at 4 °C. Red blood cells in cell pellets were lysed by ACK lysis buffer (Gibco, A1049201). Cells were stained with antibodies in 200 µl of the staining buffer (PBS with 0.5% BSA and 2 mM EDTA (Invitrogen, 15575020)) on ice for 30 min, and then washed by 2 ml of the staining buffer. Cells were resuspended in the staining buffer and analyzed using a FACS Symphony flow cytometer (BD Biosciences). DAPI was used for live/dead staining. Flowjo V10.7.1 was used for analysis. The same gating strategy was applied to flow samples from the control and Csf1$^{\Delta\Delta Adipoq}$ mice. The antibodies used for flow cytometry included anti-CD45-PerCP/Cyanine5.5 (Biolegend, clone QA17A26,1:200), anti-Ly6G-Brilliant Violet 711 (Biolegend, clone 1A8, 1:200), anti-CD11b-Alexa Fluor 700 (Biolegend, clone M1/70, 1:200), anti-F4/80- APC/Cyanine7 (Biolegend, clone BM8, 1:200), and anti-Ly6C-Brilliant Violet 510 (Biolegend, clone HK1.4, 1:200).

For BrdU staining, 10-week-old female Adipoq Cre-mTmG reporter mice were injected (i.p.) with 200 mg/kg BrdU (B5002, Sigma) 24 hr and 3 hr before sacrifice. The bone marrow cells were flushed out from long bones. The bones were then cut into small pieces and digested as described in the FACS method. The bone marrow cells and the digested bone solution were collected together, filtered through 70 µm cell strainer and spun down at 500 g for 5 min at 4 °C. Red blood cells in cell pellets were lysed by ACK lysis buffer (Gibco, A1049201). After cell viability staining (Live/Dead Fixable Blue Dead Cell Stain kit, L23105 ThermoFisher Scientific) and CD45/CD31/Ter119 cell surface antigen staining, the cells were treated using eBioscience BrdU Staining Buffer (00-5525-00) according to the manufacturer's instruction. The flowcytometry was then performed as described above with the isotype control antibody (Biolegend, Clone MOPC-21, 400119, 5 µg/sample) and the anti-BrdU antibody (Biolegend, clone 3D4, 364114, 5 µg/sample).

## Fluorescence-activated cell sorting (FACS)

Bone marrow cells from 12-week-old male Adipoq Cre-mTmG reporter mice were harvested as above. The long bones (without periosteum) were then cut into pieces and digested in α-MEM (Gibco,12561056) with 2% FBS (Atlanta Biologicals, S11550), 2 mg/ml collagenase II (Worthington, LS004176) and 1 mg/ml Dispase II (Gibco, 17105041) for 25 min in a rotary incubator at 250 rpm at 37 °C. Dnase I (2 units/ml, Sigma,4716728001) was then added. After 5 min, the digested bone solution and bone marrow cells were collected together, filtered through 70 µm cell strainer and spun down at 500 g for 5 min at 4 °C. Red blood cells in cell pellets were lysed by ACK lysis buffer (Gibco, A1049201). The cells were stained with anti-Lin antibodies, including anti-CD45-APC (Biolegend, clone 30-F11, 1:200), anti-CD31-APC (Biolegend, clone MEC13.3, 1:200) and anti-Ter119-APC (Biolegend, 116212,1:200) for 30 min at 4 °C. The Adipoq + cells (GFP+) were sorted using a FACS Aria II SORP cell sorter (Becton Dickinson) at Weill Cornell Medical College, with exclusion of DAPI+ (BD, 564907) cells, doublets and Lin+ cells.

## Isolation of peripheral mature adipocytes and bone marrow mature adipocytes

EWAT, IngWAT and brown adipose tissue (BAT) were isolated from mice and minced into small pieces by scissors and then digested by the digestion buffer (a-MEM (Gibco,12561056) with 2 mg/ml collagenase II (Worthington, LS004176) and 2% BSA (Gemini Bio, 700–100 P)) for 40 min in a rotary incubator at 250 rpm at 37 °C. The cells were then filtered through 70 µm cell strainer and spun down at 500 g for 5 min at room temperature. The floating mature lipid-laden adipocytes were collected from the top layer and washed with PBS for three times. Bone marrow cells were directly flushed out by quick centrifuge (from 0~9400 g, approximately 15 s at room temperature) after cutting both ends of long bones, then resuspended by PBS (Corning, 21040CV) and filtered through 70 µm cell strainer (FALCON,352350). The cells were spun down at 500 g for 5 min at room temperature. The floating mature lipid-laden adipocytes were collected from the top layer and washed with PBS for three times (*Fan et al., 2017*).

## Statistical analysis

Statistical analysis was performed using Graphpad Prism software. Two-tailed Student's t test was applied when there were only two groups of samples. In the case of more than two groups of samples, two-way ANOVA was used with more than two conditions. ANOVA analysis was followed by post-hoc Bonferroni's correction for multiple comparisons. $p < 0.05$ was taken as statistically significant. Data are presented as the mean $\pm$ SD as indicated in the figure legends.

## Acknowledgements

We thank Courtney Ng for critical review of the manuscript. We are grateful to Drs. Ugur Ayturk and Vincentius Suhardi from the Hospital for Special Surgery, and the lab members from Dr. Baohong Zhao's laboratory for their helpful discussions and assistance. We thank Dr. Ling Qin for sharing some mouse bone marrows for our revision. MBG holds a Career Award for Medical Scientists from the Burroughs Welcome Foundation, and a Pershing Square Sohn Prize for Young Investigators in Cancer Research. This work was supported by grants from the National Institutes of Health (AR075585 to MBG, AG045040 to JXJ, and AR068970, AR071463, AR078212 to BZ), Welch Foundation Grant (AQ-1507 to JXJ) and by support for the Rosensweig Genomics Center at the Hospital for Special Surgery from The Tow Foundation. The content of this manuscript is solely the responsibilities of the authors and does not necessarily represent the official views of the NIH.

## Additional information

### Competing interests

Jean X Jiang, Baohong Zhao: Reviewing editor, *eLife*. The other authors declare that no competing interests exist.

### Funding

| Funder | Grant reference number | Author |
|---|---|---|
| National Institutes of Health | AR078212 | Baohong Zhao |
| National Institutes of Health | AR068970 | Baohong Zhao |
| National Institutes of Health | AR071463 | Baohong Zhao |
| National Institutes of Health | AR075585 | Matthew B Greenblatt |
| National Institutes of Health | AG045040 | Jean X Jiang |
| Tow Foundation | Rosensweig Genomics Center at the Hospital for Special Surgery | Baohong Zhao |
| Welch Foundation | AQ-1507 | Jean X Jiang |

The funders had no role in study design, data collection and interpretation, or the decision to submit the work for publication.

### Author contributions

Kazuki Inoue, Data curation, Formal analysis, Investigation, Methodology, Software, Validation, Writing – review and editing, Performed most experiments for the first submission; Yongli Qin, Data curation, Formal analysis, Investigation, Methodology, Software, Validation, Writing – review and editing, Performed flowcytometry; Yuhan Xia, Data curation, Formal analysis, Investigation, Methodology, Validation, Writing – review and editing, Performed most experiments for the first submission; Jie Han, Software, Writing – review and editing, Assisted YQ for bioinformatics analysis; Ruoxi Yuan, Software, Writing – review and editing, Assisted YQ for bioinformatics analysis; Jun Sun, Methodology, Writing

– review and editing, Assisted flowcytometry and cell sorting; Ren Xu, Software, Writing – review and editing, Assisted YQ for bioinformatics analysis; Jean X Jiang, Matthew B Greenblatt, Resources, Funding acquisition, Writing – review and editing; Baohong Zhao, Conceptualization, Resources, Supervision, Funding acquisition, Writing - original draft, Writing – review and editing

### Author ORCIDs

Kazuki Inoue 🔟 http://orcid.org/0000-0001-6305-9374
Yongli Qin 🔟 http://orcid.org/0000-0001-6133-0511
Jean X Jiang 🔟 http://orcid.org/0000-0002-2185-5716
Baohong Zhao 🔟 http://orcid.org/0000-0002-1286-0919

### Ethics

All mouse experiments were approved by Institutional Animal Care and Use Committee of the Hospital for Special Surgery and Weill Cornell Medical College (protocol numbers: 2016-0001 and 0004).

### Decision letter and Author response

Decision letter https://doi.org/10.7554/eLife.82118.sa1
Author response https://doi.org/10.7554/eLife.82118.sa2

## Additional files

### Supplementary files

• MDAR checklist

### Data availability

The current manuscript does not contain sequencing data. The Source Data files for figures have been submitted.

The following previously published datasets were used:

| Author(s) | Year | Dataset title | Dataset URL | Database and Identifier |
|---|---|---|---|---|
| Dolgalev I, Tikhonova AN | 2021 | Connecting the Dots: Resolving the Bone Marrow Niche Heterogeneity | https://singlecell.broadinstitute.org/single_cell/study/SCP1248;https://osf.io/ne9vj | Broad Institute Single Cell Portal, SCP1248 |
| Wang Z, Li X, Yang J, Gong Y, Zhang H, Qiu X | 2021 | Single-cell RNA sequencing deconvolutes the in vivo heterogeneity of human bone marrow-derived mesenchymal stem cells | https://www-ncbi-nlm-nih-gov.ezproxy.med.cornell.edu/geo/query/acc.cgi?acc=GSE147287 | NCBI scRNAseq, GSE147287 |

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
