## [Editor Report]

This fundamental work advances our understanding of the function of a subpopulation of bone marrow cells as an important source of M-CSF to regulate bone remodeling. The evidence supporting the conclusion is compelling, using Adipoq-Cre-driven conditional deletion of Csf1 and the analysis of the publicly available scRNAseq data. This paper is of interest to skeletal biologists studying bone marrow stem/progenitor cells and bone remodeling.

---

## [Decision Letter]

**Decision letter after peer review:**

Thank you for submitting your article "Bone marrow AdipoQ-lineage progenitors are a major cellular source of M-CSF that dominates bone marrow macrophage development, osteoclastogenesis and bone mass" for consideration by *eLife*. Your article has been reviewed by 3 peer reviewers, and the evaluation has been overseen by a Reviewing Editor and Mone Zaidi as the Senior Editor. The following individual involved in review of your submission has agreed to reveal their identity: William P Cawthorn (Reviewer #3).

Essential revisions:

The work has potential but requires essential additional data to support the central claim that bone marrow AdipoQ-lineage progenitors are a major source of M-CSF that determines bone mass. Particularly, it is necessary for the authors to (1) test if M-CSF is also from other sources (peripheral adipose tissue, other bone cells etc.); (2) assess M-CSF protein levels using histological methods; (3) clarify the identity of AdipoQ-lineage cells; (4) assess the cortical bone properties and conduct mechanical testing of the Csf1∆AdipoQ mice; and (5) clarify if the phenotype results from the Csf1∆AdipoQ mice is exclusively from decreased M-CSF or also from the changes of other factors (eg. IL-34). In addition, the authors should consider addressing other concerns raised by the reviewers.

*Reviewer #1 (Recommendations for the authors):*

1. Abstract:

– It would be helpful if the authors could specify how they define AdipoQ-lineage progenitors in the abstract (i.e. AdipoQCre).

– "The postmenopausal osteoporosis in a mouse model" is not accurate. It should be replaced by "ovariectomized mice".

2. Introduction:

– RE: "there is no clear evidence supporting whether osteoblasts and osteocytes express M-CSF in vivo…". A published study from one of the coauthors seems to have already addressed this issue: JBMR Plus. 4:e10080.

3. Results:

– The authors should explain briefly why they designated Adipoq+ cells in scRNAseq as MSPC-adipo. Do these cells express any MSPC markers?

– The qPCR experiment shown in Figure 1E is extremely important for supporting the major conclusion. Despite the importance, the authors examined only one gene expression (Csf1). The authors should also demonstrate expression levels of other genes that may serve as a control, such as Lpl.

– They should also clarify that Csf1 was expressed approximately 4-fold higher in AdipoQ+ cells compared to other mature adipocytes.

– RE: "the bone loss extent was less in the Csf1dAdipoQ mice than in the control mice" Can the authors specify which data they refer to?

4. Discussion:

– "AdipoQ is exclusively expressed…": this sentence needs to be revised, as it is not entirely accurate.

- It would be helpful if the authors could also discuss the similarities and differences of bone phenotypes between AdipoQCre-driven M-CSF and RANKL CKO models.

*Reviewer #2 (Recommendations for the authors):*

Congratulations on making this exciting discovery from scRNA-seq datasets and validating it in animal models. The functional studies indicate that AdipoQ cells in the bone marrow produce appreciable amounts of Csf1 and this is one means by which they have an outsized effect on bone remodeling. There are many strengths to the data and presentation as listed in the public comments. I have a few questions and suggestions for improving the study and manuscript.

1) AdipoQ-Cre mice are not included as a control in osteopetrosis experiments. Do you have any phenotyping data on these animals from your colony?

2) Are there any differences in cortical bone properties? Mechanical testing of Csf1∆AdipoQ mice would help to determine if they are really osteopetrotic?

3) Are there any differences in vertebral BMD in Csf1∆AdipoQ mice?

4) The phenotyping of the Csf1∆AdipoQ mice is somewhat superficial. Does IL-34 decrease in Csf1∆AdipoQ mice? What about other cytokines and transcripts? Has a scRNA-seq or even bulk RNA-seq analysis been done on the bone marrow of Csf1∆AdipoQ mice and controls to examine transcriptomic changes other than M-CSF in these animals?

5) Does adding M-CSF back Csf1∆AdipoQ bone marrow cultures induce osteoclast formation? The assay seems to have been done in medium containing phenol red, which may have slowed differentiation and is why cultures were 10 days long. What happens to osteoclast differentiation in phenol red-free medium?

6) Do AdipoQ+ cells exist in human bone marrow? If so, do these human cells produce M-CSF?

7) There are no in vitro assays of macrophage function (e.g., LPS stimulation).

8) Show SD instead of SEM in figures.

9) Figure 1A: Please label the clusters in the UMAP plot with numbers for those who are color blind.

10) Figure 1C does not appear to be the same as that generated on the Single Cell Portal (link here), which is linked to the site, according to the methods, where these data are housed (https://osf.io/ne9vj). Specifically, chondrocytes seem to make more Csf1 than shown in Figure 1C. Please explain.

11) Figure 1E. BMA and BM-AdipoQ only have two data points. Please increase to at least three. Ideally, it would be more for all groups.

*Reviewer #3 (Recommendations for the authors):*

a. The most-important issue to address in this paper is whether the bone and macrophage phenotypes result exclusively from loss of M-CSF expression in bone marrow (BM) AdipoQ-lineage progenitors, or if decreased M-CSF from peripheral adipose depots also contributes to these phenotypes. The specific points are as follows:

1. Firstly, the authors repeatedly state that Csf1 expression in adipocytes or peripheral adipose tissues is minimal and therefore cannot contribute to their phenotype. For example:

i) Pages 4-5, "AdipoQ+ cells in peripheral adipose tissue and mature bone marrow adipocytes almost do not express M-CSF". Please can you provide evidence to support this, e.g. a reference or some direct data? As noted above, transcriptomic profiles show relatively high expression of Csf1 in mouse white adipose tissue, suggesting that such peripheral adipose tissues do contribute to systemic M-CSF.

ii) Page 5, ""Besides bone marrow, peripheral adiposes contain a large amount of AdipoQ+ mature adipocytes. These cells however do not produce Csf1 (Figure 1E)."

iii) Page 7: "peripheral AdipoQ+ adipocytes… express neither RANKL nor M-CSF (our findings and (29)). The unique expression of M-CSF and RANKL by bone marrow AdipoQ-lineage progenitors is a feature that distinguishes this cell population from other adipose depots." But this is not the case: the Fan 2017 paper does not assess M-CSF expression, while many studies demonstrate that white and brown adipocytes do express M-CSF

However, there is ample evidence showing that adipocytes and adipose tissue have high M-CSF expression and that this has biological functions, e.g.

– https://link.springer.com/article/10.1007/s12257-020-0023-8

– https://www.ncbi.nlm.nih.gov/pmc/articles/PMC508735/

– https://www.ncbi.nlm.nih.gov/geo/tools/profileGraph.cgi?ID=GDS3142:1460220_a_at

2. Secondly, qPCR is used to compare Csf1 expression between mature adipocytes and the BM AdipoQ-lineage progenitors (Figure 1E). This seems to be the basis of the authors' conclusion that there is minimal M-CSF production from peripheral adipose tissue. However, qPCR comparison across such different tissue types is fraught with difficulties and is highly dependent on using a suitable housekeeping gene. Here, Gapdh is used as a housekeeping gene; please show the expression of Gapdh across the five cell types. If expression is highly variable then this will greatly influence the normalized Csf1 values. this is expressed at a similar level across the five cell types, or is its expression very variable? It is also more informative to assess protein expression, rather than transcript expression.

Please address these issues as follows:

3. To better interpret figure 1E please include a supplemental figure showing Gapdh expression across the five cell types. I also recommend that you use several housekeeping genes (e.g. Gapdh, Tbp, Ppia, rn18s), calculate the geometric mean of their expression, and normalise Csf1 to this mean value (Vandesompele et al., 2002).

4. It is essential to compare Csf1 expression in peripheral adipose tissue vs whole bone marrow in control vs KO mice. Ideally, you would analyse Csf1 transcripts in adipocytes and stromal vascular cells from peripheral adipose tissue, to determine the cellular source of the Csf1. You would then analyse M-CSF protein in whole adipose tissue and whole bone marrow from the control vs KO mice. If both of these analyses cannot be done (transcript in fractionated tissue; protein in whole tissues) then the most-informative analysis would be for M-CSF protein in whole tissues. This would show if the KO alters overall protein expression in peripheral adipose tissue and/or in bone marrow.

b. AdipoQ-Cre cannot distinguish between MSPC-Adipo and MSPC-Osteo.

A second major point is that it is not clear if the BM AdipoQ-lineage progenitors correspond to the MSPC-Adipo cells only, or if they also include the MSPC-Osteo cells described in figure 1. Indeed, the MSPC-Osteo cells also express Adipoq (Figure 1D) and even low-level Cre expression from the Adipoq promoter will be sufficient to ablate Csf1 in these cells. This is an issue because on page 7 you conclude, "These results together with the scRNAseq data in Figure 1 demonstrate that AdipoQ-lineage progenitor cells in bone marrow produce substantially more M-CSF than osteoblast lineage cells", while in the results (page 4) you state, "Given that bone marrow AdipoQ-lineage progenitors constitute only about 0.08% of bone marrow cells (Suppl. Figure 1), these results support that bone marrow AdipoQ-lineage progenitors are the major cellular source of M-CSF expression in the bone marrow." However, if Csf1 is also deleted in many of the MSPC-Osteo cells then you can't draw this conclusion. Are you able to check if the Csf1-deltaAdipoQ model maintains Csf1 expression in the MSPC-Osteo cells while ablating Csf1 in the MSPC-Adipo cells? If this cannot be assessed experimentally then please update the Discussion to clearly state this limitation of the model. Please also update Supplemental Figure 1 to show the % of MSPC-Osteo cells in the bone marrow (if this % is very high then it adds to the concern that deletion of Csf1 in the MSPC-Osteo cells may be influencing the decrease in total Csf1 expression in BMSCs).

References

Berry, R., and Rodeheffer, M.S. (2013). Characterization of the adipocyte cellular lineage in vivo. Nat Cell Biol 15, 302-308. 10.1038/ncb2696.

Bravenboer, N., Bredella, M.A., Chauveau, C., Corsi, A., Douni, E., Ferris, W.F., Riminucci, M., Robey, P.G., Rojas-Sutterlin, S., Rosen, C., et al. (2020). Standardised Nomenclature, Abbreviations, and Units for the Study of Bone Marrow Adiposity: Report of the Nomenclature Working Group of the International Bone Marrow Adiposity Society. Front. Endocrinol. (Lausanne) 10. 10.3389/fendo.2019.00923.

Vandesompele, J., De Preter, K., Pattyn, F., Poppe, B., Van Roy, N., De Paepe, A., and Speleman, F. (2002). Accurate normalization of real-time quantitative RT-PCR data by geometric averaging of multiple internal control genes. Genome biology 3, RESEARCH0034.

[Editors’ note: further revisions were suggested prior to acceptance, as described below.]

Thank you for resubmitting your work entitled "Bone marrow Adipoq-lineage progenitors are a major cellular source of M-CSF that dominates bone marrow macrophage development, osteoclastogenesis and bone mass" for further consideration by *eLife*. Your revised article has been evaluated by Mone Zaidi (Senior Editor) and a Reviewing Editor.

The manuscript has been improved but there are some remaining issues that need to be addressed, as outlined in the following reviewers' comments.

*Reviewer #1 (Recommendations for the authors):*

In this revised manuscript, the authors carefully addressed the concerns raised by the three reviewers with additional data. The manuscript is now well-written, and the discussion is particularly thorough. My remaining requests are listed below.

1. Abstract, Line 56-58: It is a bit off-the-mark and awkward to assert it in the summary sentence that "Here, we identify bone marrow Adipoq-lineage progenitors, which are…". This has been done in a previous study and is not the major achievement of the current study. This sentence should be shuffled with the subsequent sentence.

2. Results, Line 193: "M-CSF is undetectable in mature adipocytes" – the authors' data (now 20-fold enrichment) does not support the absence of M-CSF in mature adipocytes. They should rephrase "undetectable" in this subtitle (nearly undetectable? negligibly expressed?).

*Reviewer #2 (Recommendations for the authors):*

The authors have answered my questions adequately. I have no further concerns and recommend acceptance.

*Reviewer #3 (Recommendations for the authors):*

Overall, I think the authors have done a very thorough job of addressing the comments from the previous reviews. The inclusion of scRNAseq data from human bone marrow is particularly welcome and informative, as is the analysis of M-CSF protein in bone marrow of the control and CKO mice. I and the other two reviewers also questioned the identity of the MSPC-Adipo and MSPC-Osteo populations and whether Adipoq expression can sufficiently distinguish these. Here, the authors have now convincingly explained why Adipoq is a suitably robust marker for the MSPC-Adipo population.

Despite these very helpful and informative responses, the following issues remain to be adequately addressed:

1. Does the CKO phenotype result only from a lack of M-CSF production from BM Adipoq-lineage progenitors, or do other cells contribute to this phenotype?

This was raised by Reviewer 1 (point 1) and me (Reviewer 3, point 1). One concern is that, because WAT has high expression of M-CSF (Csf1), and because Adipoq is expressed in WAT, the Adipoq-Cre may delete Csf1 in WAT and this may contribute to the CKO phenotype. The authors have tried to address this in several ways, including analysis of M-CSF protein in WAT adipocytes and SVF from control mice, and updating their qPCR analysis of Csf1 expression across various cell types (Figure 3A and 3D in the revised manuscript). These updates are helpful and demonstrate clearly that M-CSF is produced from SVF, but not adipocytes, of WAT. Together with previous data showing that Adipoq-Cre does not target WAT SVF, these data suggest that the CKO model will not have altered M-CSF expression in WAT. However, the way to address this, as requested in the previous reviews, is to measure the expression of Csf1 transcripts and/or M-CSF protein in the adipocyte and SVF fractions of WAT from control vs CKO mice. The authors have not done this, and therefore it remains to be conclusively proven that the CKO phenotype results exclusively from M-CSF depletion in the BM Adipoq-lineage progenitor cells. I agree that this is the likely explanation, but the contribution of WAT SVF cells cannot be conclusively ruled out. Therefore, I think the authors must either measure Csf1 transcripts (and/or M-CSF protein) in the WAT SVF of the control vs CKO mice, thereby fully addressing this possibility; or they should update the Discussion clearly stating that this possibility is one remaining limitation of the present study, however unlikely it may be.

2. Related to the previous point, the authors have updated the qPCR analysis (now Figure 3A) by using both Gapdh and Actb as housekeeping genes. However, I didn't see Actb data in the source data file. Please can this be included? The source data does show that Gapdh is relatively consistent across the cell types, but its expression is still lowest in the BM Adipoq-lineage progenitors and therefore this lower housekeeping gene expression will be increasing the apparent Csf1 expression in these cells. In addition, this figure compares the BM Adipoq-lineage progenitors only to adipocytes; it would be more informative to compare the expression to WAT SVF since we now know that it is the SVF, not the adipocytes, that is the source of M-CSF in WAT.

3. Contrary to the authors' conclusions, their data show that the CKO does not alter the effect of OVX on bone loss (Figure 7). Both I (point k.) and Reviewer 1 (point 4) questioned the conclusions from the OVX study. The data for this clearly show that the genotype influences bone parameters in both sham and OVX mice, but it does not obviously alter the OVX effect within each genotype. The authors have tried to address this by showing the % change that OVX causes for each parameter, in each genotype (Figure 7C). This is not illogical, but it is also not statistically informative and therefore doesn't address our original critique. I asked the authors to do a 2-way ANOVA and test if there is a significant "genotype x OVX" interaction, i.e. does the CKO alter the OVX effect? This was a very straightforward request, but the authors have not done it. Thankfully, they have provided the source data, so I decided to do the analysis myself (my analysis file (Prism) can be downloaded at https://www.dropbox.com/t/WAZb3y34d8p4Skzn). This analysis shows that there is a genotype x OVX interaction for trabecular spacing (Tb.Sp), but not for any of the other parameters tested. I decided to do a further 3-way ANOVA, across all five of the parameters tested (normalizing each to the average for each parameter) to give even more power to detect any genotype x OVX interactions. This reveals a "genotype x OVX" P value of 0.947, which demonstrates conclusively that there is no overall effect of the CKO on OVX-induced bone loss.

This conclusion is robust and does not undermine interest in the manuscript: it isn't essential that the BM Adipoq-lineage-derived M-CSF contributes to OVX-induced bone loss, so the authors shouldn't feel compelled to make this conclusion when the data don't support it. Indeed, it is interesting that BM Adipoq-lineage-derived M-CSF may influence bone remodelling in some circumstances but not in others.

---

## [Author Response]

Essential revisions:The work has potential but requires essential additional data to support the central claim that bone marrow AdipoQ-lineage progenitors are a major source of M-CSF that determines bone mass. Particularly, it is necessary for the authors to (1) test if M-CSF is also from other sources (peripheral adipose tissue, other bone cells etc.); (2) assess M-CSF protein levels using histological methods; (3) clarify the identity of AdipoQ-lineage cells; (4) assess the cortical bone properties and conduct mechanical testing of the Csf1∆AdipoQ mice; and (5) clarify if the phenotype results from the Csf1∆AdipoQ mice is exclusively from decreased M-CSF or also from the changes of other factors (eg. IL-34). In addition, the authors should consider addressing other concerns raised by the reviewers.

We have addressed each of these essential concerns (1-5) in the Response to Reviewers letter and in the revised manuscript. Specifically,

1) Test if M-CSF is also from other sources (peripheral adipose tissue, other bone cells etc.)

Please refer to the answers to Reviewer 1 Q1; Reviewer 3 Weakness Q1, Q2, Major Qa.

2) Assess M-CSF protein levels using histological methods

Please refer to the answers to Reviewer 1 Q1, Q2; Reviewer 3 Weakness Q2, Major Qa.

3) Clarify the identity of AdipoQ-lineage cells

Please refer to the answers to Reviewer 1 Q3, Abstract/Q1, Results/Q1; Reviewer 3 Weakness Q3, Major Qb.

4) Assess the cortical bone properties and conduct mechanical testing of the Csf1∆AdipoQ mice

Please refer to the answers to Reviewer 1 Q2; Reviewer 2 Q2.

5) Clarify if the phenotype results from the Csf1∆AdipoQ mice is exclusively from decreased M-CSF or also from the changes of other factors (eg. IL-34)

Please refer to the answers to Reviewer 2 Q4, Q5.

Reviewer #1 (Recommendations for the authors):1. Abstract:– It would be helpful if the authors could specify how they define AdipoQ-lineage progenitors in the abstract (i.e. AdipoQCre).

We edited the sentence as ‘we identify bone marrow Adipoq-lineage progenitors, which are characterized by a low level of proliferation, and high expression of *Adipoq* and bone marrow stromal cell markers’ in the Abstract.

– "The postmenopausal osteoporosis in a mouse model" is not accurate. It should be replaced by "ovariectomized mice".

We corrected this description as suggested.

2. Introduction:– RE: "there is no clear evidence supporting whether osteoblasts and osteocytes express M-CSF in vivo…". A published study from one of the coauthors seems to have already addressed this issue: JBMR Plus. 4:e10080.

We corrected this description with the JBMR Plus paper cited as suggested.

3. Results:– The authors should explain briefly why they designated Adipoq+ cells in scRNAseq as MSPC-adipo. Do these cells express any MSPC markers?

Yes, these cells express high levels of MSPC markers, such as Lepr, Cxcl12 and Kitl. We have added a paragraph to explain on pg. 4: ‘Since we utilized Adipoq Cre mice to investigate the function of this progenitor population, we used the nomenclature bone marrow Adipoq-lineage progenitors to designate these cells throughout this study. We found the Adipoq-lineage progenitors to be highly enriched for bone marrow stromal cell markers important for the hematopoietic niche, such as *Lepr*, *Cxcl12* and *Kitl*, but also express unique genes, such as *Lpl* (Figure 1B). These cells labeled by Adipoq Cre were relatively quiescent, with about 4% cells incorporating BrdU under basal conditions (Figure 1C). The Adipoq-lineage progenitors are not mature adipocytes, but express some common adipocyte lineage markers, such as *Cebpa* and *Adipoq* (Figure 1B). On the other hand, osteoblast lineage marker genes, such as *Sp7*, *Alpl*, *Dmp1* and *Bglap*, are nearly undetectable in these cells (Figure 1B).’

– The qPCR experiment shown in Figure 1E is extremely important for supporting the major conclusion. Despite the importance, the authors examined only one gene expression (Csf1). The authors should also demonstrate expression levels of other genes that may serve as a control, such as Lpl.

Following the reviewer’s suggestion, we have added data showing *Lpl* expression in Figure 3A (old Fig1E). Lpl is highly expressed in the bone marrow AdipoQ lineage progenitors, which is also found in scRNAseq results (Figure 1B, Figure 2B, ref.22), but almost undetectable in bone marrow mature adipocytes. Lpl is expressed in peripheral white adipocytes, which is known knowledge in adipocyte field.

– They should also clarify that Csf1 was expressed approximately 4-fold higher in AdipoQ+ cells compared to other mature adipocytes.

We thank the reviewer for bringing out this important question. Reviewer 3 suggested us to compare Csf1 based on evenly expressed Gapdh or geometric mean of housekeeping genes when using samples from different tissues. We followed this instruction, and happily found that the results turned out even better, with 20-30-fold higher in AdipoQ lineage progenitor cells than in other mature adipocytes. We appreciate both reviewers’ comments, which helped us obtain more robust results. The data are now shown in Figure 3A and Figure 3-Suppl Figure 1.

– RE: "the bone loss extent was less in the Csf1dAdipoQ mice than in the control mice" Can the authors specify which data they refer to?

We mainly refer to the decrease in BV/TV and changes in several other μCT parameters by OVX as the extent of bone loss. The decrease in BV/TV after OVX: 35.1% in WT vs 20.9% in *Csf1^∆AdipoQ^* mice shown in Figure 7C (old Figure 5C) indicates that the extent of bone loss was less in the *Csf1^∆AdipoQ^* mice than in the control mice. The value changes, including those from other μCT parameters, have been included in Figure 7C.

4. Discussion:– "AdipoQ is exclusively expressed…": this sentence needs to be revised, as it is not entirely accurate.

Yes, we removed ‘exclusively’ as suggested.

- It would be helpful if the authors could also discuss the similarities and differences of bone phenotypes between AdipoQCre-driven M-CSF and RANKL CKO models.

Compared to the findings in Dr. Ling Qin group’s JCI paper, the osteopetrosis phenotype in long bones are similar between AdipoQCre-driven M-CSF and RANKL CKO mice. Interestingly, AdipoQ-Cre driven RANKL cKO mice also exhibit osteopetrosis in lumbar, while *Csf1^∆AdipoQ^* mice do not (Figure 4—figure supplement 5). The mechanisms underlying this difference are unclear. A possibility is that cells other than AdipoQ-expressing cells in lumbar presumably produce sufficient M-CSF or IL34 that could compensate the loss of M-CSF by AdipoQ-expressing cells. This difference also implicates the presence of distinct cellular compartments and microenvironments between long bones and vertebral bones. These discussions have been added to the Discussion section on pg.8.

Reviewer #2 (Recommendations for the authors):Congratulations on making this exciting discovery from scRNA-seq datasets and validating it in animal models. The functional studies indicate that AdipoQ cells in the bone marrow produce appreciable amounts of Csf1 and this is one means by which they have an outsized effect on bone remodeling. There are many strengths to the data and presentation as listed in the public comments. I have a few questions and suggestions for improving the study and manuscript.1) AdipoQ-Cre mice are not included as a control in osteopetrosis experiments. Do you have any phenotyping data on these animals from your colony?

We compared the bone phenotype between *Csf1^f/f^* and *Csf1^f/+^;AdipoQ* mice, but did not find significant difference. The results are shown in Figure 4—figure supplement 4.

2) Are there any differences in cortical bone properties? Mechanical testing of Csf1∆AdipoQ mice would help to determine if they are really osteopetrotic?

*Csf1^∆AdipoQ^* mice do not exhibit abnormal cortical bone phenotype. The cortical bone parameters are now included in Figure 4G. The bone samples were stored in ethanol. Mechanical testing using the fixed samples may not reflect natural bone properties of each sample, but could be used for comparison of the bone strength between samples. We also consulted with our biomechanical core at HSS who supported the comparison. As shown in Author response image 1, there is no significant mechanical difference between the WT control and *Csf1^∆AdipoQ^* long bones using 4-point bending test.

**Author response image 1. sa2fig1:** 

3) Are there any differences in vertebral BMD in Csf1∆AdipoQ mice?

There are no differences in vertebral BMD or bone mass between the WT control and *Csf1^∆AdipoQ^* mice. The results are shown in Figure 4—figure supplement 5. The mechanisms underlying this phenomenon are unclear. A possibility is that cells other than AdipoQ-expressing cells in lumbar presumably produce sufficient M-CSF or IL34 that could compensate the loss of M-CSF by AdipoQ-expressing cells in lumbar vertebrae. This difference also implicates the presence of distinct cellular compartments and microenvironments between long bones and vertebral bones. These discussions have been added to the Discussion section on pg. 8.

4) The phenotyping of the Csf1∆AdipoQ mice is somewhat superficial. Does IL-34 decrease in Csf1∆AdipoQ mice? What about other cytokines and transcripts? Has a scRNA-seq or even bulk RNA-seq analysis been done on the bone marrow of Csf1∆AdipoQ mice and controls to examine transcriptomic changes other than M-CSF in these animals?

We examined the expression of a group of cytokines that often regulate macrophage function and osteoclastogenesis in bone marrow, including Il34, Il1b, Il6, Il10, Csf2, TNF, Cxcl12, and found that the deficiency of Csf1 in *Csf1^∆AdipoQ^* mice did not affect the expression of these genes. The results are shown in Figure 4-Suppl Figure 2.

5) Does adding M-CSF back Csf1∆AdipoQ bone marrow cultures induce osteoclast formation? The assay seems to have been done in medium containing phenol red, which may have slowed differentiation and is why cultures were 10 days long. What happens to osteoclast differentiation in phenol red-free medium?

Yes, when we added recombinant M-CSF back to the *Csf1^∆AdipoQ^* cell cultures, the osteoclast formation is similar to that in the WT control cultures. The results are now shown in Figure 5E. We have been using medium containing phenol red for years as a quick and convenient monitor for general health of cell cultures. General osteoclastogenic cultures in our lab usually take 4-5 days but not 10 days to obtain giant osteoclasts because exogenous M-CSF (20ng/ml) and RANKL are both added to the cultures, which facilitate the differentiation process. However, in this paper, we wanted to test if endogenous MCSF, which level is lower than that of exogenous M-CSF we usually use for culture, is able to induce osteoclastogenesis. The WT cell cultures took 10 days to differentiate to mature osteoclasts because exogenous M-CSF was not added. The results showing no osteoclasts formed in the C*sf1^∆AdipoQ^* cell cultures corroborate the importance of AdipoQ lineage progenitors’ contribution to M-CSF level.

6) Do AdipoQ+ cells exist in human bone marrow? If so, do these human cells produce M-CSF?

We thank the reviewer for this insightful question. Yes, we analyzed scRNAseq datasets from human bone marrow samples from a published article and a BioRxiv manuscript. Both datasets show the existence of the AdipoQ lineage progenitors that express highest level of *Csf1* among the analyzed cell clusters. The results are shown in Figure 2 and Figure 2—figure supplement 1, and discussed on pg. 4.

7) There are no in vitro assays of macrophage function (e.g., LPS stimulation).

We stimulated bone marrow derived macrophages (BMMs) with LPS and found that the inflammatory response of BMMs, as indicated by the inflammatory gene induction (Figure 5—figure supplement Figure 2A) and the activation of MAPK or NF-κB pathways in response to LPS (Figure 5—figure supplement Figure 2B), was similar between the BMMs derived from control and C*sf1^∆AdipoQ^* mice. The results are shown in Figure 5-Suppl Figure 2.

8) Show SD instead of SEM in figures.

Yes, we have changed SEM to SD, and this change does not affect results and conclusions.

9) Figure 1A: Please label the clusters in the UMAP plot with numbers for those who are color blind.

Yes, we have now labeled clusters with numbers.

10) Figure 1C does not appear to be the same as that generated on the Single Cell Portal (link here), which is linked to the site, according to the methods, where these data are housed (https://osf.io/ne9vj). Specifically, chondrocytes seem to make more Csf1 than shown in Figure 1C. Please explain.

We double checked the data shown in the Signal Cell Portal, and Author response image 2 is a screen shot of the Csf1 expression from the site. As shown in this screen shot, only 3.85% chondrocytes are positive with a very low expression level of Csf1 expression at 0.04 scaled mean expression, which is much less than AdipoQ lineage progenitor cells (MSPC-Adipo in this Portal).

The legends of dot sizes and scaled mean expression bar in the Portal are different from our figures, which were generated via R and Seurat. We have also provided the source data extracted from the Frontier paper (ref. 27) for the cell numbers in each cluster, % expressed cells and scaled average expression values for Figure 1.

11) Figure 1E. BMA and BM-AdipoQ only have two data points. Please increase to at least three. Ideally, it would be more for all groups.

Yes, we have increased to 5 replicates for each group.

Reviewer #3 (Recommendations for the authors):a. The most-important issue to address in this paper is whether the bone and macrophage phenotypes result exclusively from loss of M-CSF expression in bone marrow (BM) AdipoQ-lineage progenitors, or if decreased M-CSF from peripheral adipose depots also contributes to these phenotypes. The specific points are as follows:1. Firstly, the authors repeatedly state that Csf1 expression in adipocytes or peripheral adipose tissues is minimal and therefore cannot contribute to their phenotype. For example:i) Pages 4-5, "AdipoQ+ cells in peripheral adipose tissue and mature bone marrow adipocytes almost do not express M-CSF". Please can you provide evidence to support this, e.g. a reference or some direct data? As noted above, transcriptomic profiles show relatively high expression of Csf1 in mouse white adipose tissue, suggesting that such peripheral adipose tissues do contribute to systemic M-CSF.ii) Page 5, ""Besides bone marrow, peripheral adiposes contain a large amount of AdipoQ+ mature adipocytes. These cells however do not produce Csf1 (Figure 1E)."iii) Page 7: "peripheral AdipoQ+ adipocytes… express neither RANKL nor M-CSF (our findings and (29)). The unique expression of M-CSF and RANKL by bone marrow AdipoQ-lineage progenitors is a feature that distinguishes this cell population from other adipose depots." But this is not the case: the Fan 2017 paper does not assess M-CSF expression, while many studies demonstrate that white and brown adipocytes do express M-CSFHowever, there is ample evidence showing that adipocytes and adipose tissue have high M-CSF expression and that this has biological functions, e.g.– https://link.springer.com/article/10.1007/s12257-020-0023-8– https://www.ncbi.nlm.nih.gov/pmc/articles/PMC508735/– https://www.ncbi.nlm.nih.gov/geo/tools/profileGraph.cgi?ID=GDS3142:1460220_a_at

Answers to (i) and (ii): We performed western blot to analyze M-CSF protein expression in peripheral adipose. As shown in Figure 3D, the stromal vascular fraction (SVF) cells in adipose, which contain multiple cell populations including adipogenic progenitors, express M-CSF. On the contrary, M-CSF was nearly undetectable in the peripheral mature adipocytes isolated from adipose (Figure 3D). These data collectively support that mature adipocytes are not a significant source of M-CSF as evidenced by nearly undetectable M-CSF expression compared to the Adipoq-lineage progenitors. However, we understand that current techniques may have limitation in identification of trace amount of M-CSF. We thus deleted descriptions such as ‘exclusive’ or ‘do not produce/express…’ in the revised manuscript.

Answers to (iii): We should have separated the citations. Fan 2017 paper tested RANKL, and our manuscript tested M-CSF. These have been corrected in the revised manuscript.

Adipose as a tissue contains many different cells and does express considerable amount of M-CSF. However, the M-CSF is mainly expressed by the SVF in the adipose tissue, but not by mature adipocytes, which is clearly supported by our Figure 3A and 3D. What we claimed in our manuscript is that mature adipocytes are not a significant source of M-CSF as evidenced by nearly undetectable MCSF expression compared to the Adipoq-lineage progenitors. We did not mention that adipose tissue does not produce M-CSF.

Furthermore, the results from the links do not support that adipocytes express a significant amount of M-CSF. The data from these links show adipose tissue or 3T3-L1 cell line cultures express M-CSF.

– *https://link.springer.com/article/10.1007/s12257-020-0023-8* M-CSF expression shown in Adipose tissue (Figure 1A, B, Figure 4A), 3T3-L1 cell line cultures (Figure 1C, Figure 6) and in vitro differentiation cultures (Figure 4B) in this paper (Biotechnol Bioproc E 25, 29–38 (2020).)

– *https://www.ncbi.nlm.nih.gov/pmc/articles/PMC508735/* The IHC staining results, to our view, showed that M-CSF was expressed in SVF in adipose. The western blot for M-CSF does not have a positive or negative control showing that M-CSF expressed samples are AdipoQ positive or not.

– *https://www.ncbi.nlm.nih.gov/geo/tools/profileGraph.cgi?ID=GDS3142:1460220_a_at* This dataset shows that it is Adipose Tissue that expresses M-CSF, not mature adipocytes.

2. Secondly, qPCR is used to compare Csf1 expression between mature adipocytes and the BM AdipoQ-lineage progenitors (Figure 1E). This seems to be the basis of the authors' conclusion that there is minimal M-CSF production from peripheral adipose tissue. However, qPCR comparison across such different tissue types is fraught with difficulties and is highly dependent on using a suitable housekeeping gene. Here, Gapdh is used as a housekeeping gene; please show the expression of Gapdh across the five cell types. If expression is highly variable then this will greatly influence the normalized Csf1 values. this is expressed at a similar level across the five cell types, or is its expression very variable? It is also more informative to assess protein expression, rather than transcript expression.

We increased the replicates of each group cells in Figure 3A (the old Figure 1E) to five/group. The GAPDH expression is pretty similar among samples (Source data of the cycles provided along with the manuscript).

We have examined M-CSF at protein levels; please refer to the answer to Q1 and Figure 3B, C, D.

Please address these issues as follows:3. To better interpret figure 1E please include a supplemental figure showing Gapdh expression across the five cell types. I also recommend that you use several housekeeping genes (e.g. Gapdh, Tbp, Ppia, rn18s), calculate the geometric mean of their expression, and normalise Csf1 to this mean value (Vandesompele et al., 2002).

The GAPDH expression is pretty similar among samples (Source data of the cycles provided along with the manuscript).

According to the reviewer’s recommendation, we used geometric mean of Gapdh and Actb expression, and happily found that the results of Csf1 expression turned out even better, with 20-fold higher in AdipoQ lineage progenitor cells than in other mature adipocytes. We appreciate the reviewer’ recommendation that helped us obtain more robust results. The data are now shown in Figure 3A and Figure 3-Suppl Figure 1.

4. It is essential to compare Csf1 expression in peripheral adipose tissue vs whole bone marrow in control vs KO mice. Ideally, you would analyse Csf1 transcripts in adipocytes and stromal vascular cells from peripheral adipose tissue, to determine the cellular source of the Csf1. You would then analyse M-CSF protein in whole adipose tissue and whole bone marrow from the control vs KO mice. If both of these analyses cannot be done (transcript in fractionated tissue; protein in whole tissues) then the most-informative analysis would be for M-CSF protein in whole tissues. This would show if the KO alters overall protein expression in peripheral adipose tissue and/or in bone marrow.

As addressed to Q1, Q2, we have examined M-CSF at protein levels; please also refer to the answers to Q1, Q2 and Figure 3B, C, D. Basically, M-CSF was nearly undetectable in the peripheral mature adipocytes isolated from adipose (Figure 3D), or mature adipocytes in bone marrow (Figure 3C). M-CSF is expressed in SVF, but AdipoQ is not (Figure 3D). We also showed a drastic decrease in M-CSF protein expression in bone marrow AdipoQ+ cells in C*sf1^∆AdipoQ^* mice compared to the WT control mice. The results are shown in Figure 4B.

b. AdipoQ-Cre cannot distinguish between MSPC-Adipo and MSPC-Osteo.A second major point is that it is not clear if the BM AdipoQ-lineage progenitors correspond to the MSPC-Adipo cells only, or if they also include the MSPC-Osteo cells described in figure 1. Indeed, the MSPC-Osteo cells also express Adipoq (Figure 1D) and even low-level Cre expression from the Adipoq promoter will be sufficient to ablate Csf1 in these cells. This is an issue because on page 7 you conclude, "These results together with the scRNAseq data in Figure 1 demonstrate that AdipoQ-lineage progenitor cells in bone marrow produce substantially more M-CSF than osteoblast lineage cells", while in the results (page 4) you state, "Given that bone marrow AdipoQ-lineage progenitors constitute only about 0.08% of bone marrow cells (Suppl. Figure 1), these results support that bone marrow AdipoQ-lineage progenitors are the major cellular source of M-CSF expression in the bone marrow." However, if Csf1 is also deleted in many of the MSPC-Osteo cells then you can't draw this conclusion. Are you able to check if the Csf1-deltaAdipoQ model maintains Csf1 expression in the MSPC-Osteo cells while ablating Csf1 in the MSPC-Adipo cells? If this cannot be assessed experimentally then please update the Discussion to clearly state this limitation of the model. Please also update Supplemental Figure 1 to show the % of MSPC-Osteo cells in the bone marrow (if this % is very high then it adds to the concern that deletion of Csf1 in the MSPC-Osteo cells may be influencing the decrease in total Csf1 expression in BMSCs).

The molecular features between MSPC-Adipo and MSPC-osteo are distinct (Figure 1). The MSPC-Adipo cluster (AdipoQ-lineage progenitors) has 6441 cells with the highest level (scaled mean expression level at 3.01) of AdipoQ among bone marrow. In contrast, the MSPC-osteo cluster consists of 2247 cells with a very low AdipoQ expression level (scaled mean expression level at 0.68). Taken together with both average expression level and cell numbers in each cluster, the relative overall contribution to AdipoQ expression by MSPC-osteo vs MSPC-Adipo is 7.8% ((2247 x 0.68)/(6441 x 3.01)). Therefore, the expression of AdipoQ in MSPC-osteo cluster is marginal compared to that in MSPC-Adipo. The major AdipoQ Cre driven cells are MSPC-adipo in bone marrow. These points have been added to Discussion on pg. 10. [Editors’ note: further revisions were suggested prior to acceptance, as described below.]

The manuscript has been improved but there are some remaining issues that need to be addressed, as outlined in the following reviewers' comments.Reviewer #1 (Recommendations for the authors):In this revised manuscript, the authors carefully addressed the concerns raised by the three reviewers with additional data. The manuscript is now well-written, and the discussion is particularly thorough. My remaining requests are listed below.1. Abstract, Line 56-58: It is a bit off-the-mark and awkward to assert it in the summary sentence that "Here, we identify bone marrow Adipoq-lineage progenitors, which are…". This has been done in a previous study and is not the major achievement of the current study. This sentence should be shuffled with the subsequent sentence.

We agree with the reviewer, and have deleted this sentence.

2. Results, Line 193: "M-CSF is undetectable in mature adipocytes" – the authors' data (now 20-fold enrichment) does not support the absence of M-CSF in mature adipocytes. They should rephrase "undetectable" in this subtitle (nearly undetectable? negligibly expressed?).

We have added ‘nearly’ as suggested.

Reviewer #3 (Recommendations for the authors):Overall, I think the authors have done a very thorough job of addressing the comments from the previous reviews. The inclusion of scRNAseq data from human bone marrow is particularly welcome and informative, as is the analysis of M-CSF protein in bone marrow of the control and CKO mice. I and the other two reviewers also questioned the identity of the MSPC-Adipo and MSPC-Osteo populations and whether Adipoq expression can sufficiently distinguish these. Here, the authors have now convincingly explained why Adipoq is a suitably robust marker for the MSPC-Adipo population.Despite these very helpful and informative responses, the following issues remain to be adequately addressed:1. Does the CKO phenotype result only from a lack of M-CSF production from BM Adipoq-lineage progenitors, or do other cells contribute to this phenotype?This was raised by Reviewer 1 (point 1) and me (Reviewer 3, point 1). One concern is that, because WAT has high expression of M-CSF (Csf1), and because Adipoq is expressed in WAT, the Adipoq-Cre may delete Csf1 in WAT and this may contribute to the CKO phenotype. The authors have tried to address this in several ways, including analysis of M-CSF protein in WAT adipocytes and SVF from control mice, and updating their qPCR analysis of Csf1 expression across various cell types (Figure 3A and 3D in the revised manuscript). These updates are helpful and demonstrate clearly that M-CSF is produced from SVF, but not adipocytes, of WAT. Together with previous data showing that Adipoq-Cre does not target WAT SVF, these data suggest that the CKO model will not have altered M-CSF expression in WAT. However, the way to address this, as requested in the previous reviews, is to measure the expression of Csf1 transcripts and/or M-CSF protein in the adipocyte and SVF fractions of WAT from control vs CKO mice. The authors have not done this, and therefore it remains to be conclusively proven that the CKO phenotype results exclusively from M-CSF depletion in the BM Adipoq-lineage progenitor cells. I agree that this is the likely explanation, but the contribution of WAT SVF cells cannot be conclusively ruled out. Therefore, I think the authors must either measure Csf1 transcripts (and/or M-CSF protein) in the WAT SVF of the control vs CKO mice, thereby fully addressing this possibility; or they should update the Discussion clearly stating that this possibility is one remaining limitation of the present study, however unlikely it may be.

We thank the reviewer for raising this relevant point. Indeed, we have performed the requested experiment of measuring Csf1 expression in the SVF, finding no evidence of alteration in Csf1f/f;adipoq-cre mice (new Figure 4-Suppl Figure 2). We note our additional data discussed by the reviewer and provided in Figure 3D demonstrating that WAT adipocytes do not express Csf1/M-CSF (Figure 3D). These results are consistent with the additional data we added demonstrating that adipoq-cre does not target SVF cells (new Figure 4-Suppl Figure 2).

2. Related to the previous point, the authors have updated the qPCR analysis (now Figure 3A) by using both Gapdh and Actb as housekeeping genes. However, I didn't see Actb data in the source data file. Please can this be included? The source data does show that Gapdh is relatively consistent across the cell types, but its expression is still lowest in the BM Adipoq-lineage progenitors and therefore this lower housekeeping gene expression will be increasing the apparent Csf1 expression in these cells. In addition, this figure compares the BM Adipoq-lineage progenitors only to adipocytes; it would be more informative to compare the expression to WAT SVF since we now know that it is the SVF, not the adipocytes, that is the source of M-CSF in WAT.

Thanks for this question. The Actb data has been provided in ‘Figure 3—figure supplement 1’ source data. The expression of Actb is also relatively consistent across the cell types.

We thank the reviewer for the suggestion to compare expression of Csf1 in BM adipoq-lineage progenitors to WAT SVF, and we have added additional data to Figure 3A showing that WAT SVF expresses Csf1, the levels of which are lower than those expressed in the BM adipoq-lineage progenitors.

3. Contrary to the authors' conclusions, their data show that the CKO does not alter the effect of OVX on bone loss (Figure 7). Both I (point k.) and Reviewer 1 (point 4) questioned the conclusions from the OVX study. The data for this clearly show that the genotype influences bone parameters in both sham and OVX mice, but it does not obviously alter the OVX effect within each genotype. The authors have tried to address this by showing the % change that OVX causes for each parameter, in each genotype (Figure 7C). This is not illogical, but it is also not statistically informative and therefore doesn't address our original critique. I asked the authors to do a 2-way ANOVA and test if there is a significant "genotype x OVX" interaction, i.e. does the CKO alter the OVX effect? This was a very straightforward request, but the authors have not done it. Thankfully, they have provided the source data, so I decided to do the analysis myself (my analysis file (Prism) can be downloaded at https://www.dropbox.com/t/WAZb3y34d8p4Skzn). This analysis shows that there is a genotype x OVX interaction for trabecular spacing (Tb.Sp), but not for any of the other parameters tested. I decided to do a further 3-way ANOVA, across all five of the parameters tested (normalizing each to the average for each parameter) to give even more power to detect any genotype x OVX interactions. This reveals a "genotype x OVX" P value of 0.947, which demonstrates conclusively that there is no overall effect of the CKO on OVX-induced bone loss.This conclusion is robust and does not undermine interest in the manuscript: it isn't essential that the BM Adipoq-lineage-derived M-CSF contributes to OVX-induced bone loss, so the authors shouldn't feel compelled to make this conclusion when the data don't support it. Indeed, it is interesting that BM Adipoq-lineage-derived M-CSF may influence bone remodelling in some circumstances but not in others.

We thank the reviewer for this comment, which is well taken. In OVX induced osteoporosis model, we found that the extent of bone loss in Csf1^∆Adipoq^ mice was less than that of the control mice, and the bone mass was significantly higher in Csf1^∆Adipoq^ mice than control mice after OVX. These results indicate a significant role for the M-CSF produced by Adipoq+ cells in estrogen-deficiency induced bone loss. Interestingly, however, as the reviewer commented, no genotype by OVX interaction term was seen in two-way ANOVA, even though there is a significant difference in the post-OVX bone mass between genotypes. The difference in baseline bone mass pre-OVX between genotypes could presumably complicate and compromise the significance analysis of genotype by OVX interaction. Further study, for example, using inducible Adipoq-cre mice, is needed to compare whether Adipoqlineage progenitors-derived M-CSF specifically regulates OVX responses or pathological bone loss induced by other stimuli, or is instead more involved in basal homeostasis. These have been added to Discussion on pg. 10.